# Exploring validation metrics for offline model-based optimisation with diffusion models

**Christopher Beckham** *first.last@mila.quebec*
*Mila, Polytechnique Montréal, ServiceNow Research†*
**Alexandre Piché** *first.last@servicenow.com*
*Mila, Université de Montréal, ServiceNow Research*
**David Vazquez** *first.last@servicenow.com*
*ServiceNow Research*
**Christopher Pal** *first.last@polymtl.ca*
*Mila, Polytechnique Montréal, ServiceNow Research, CIFAR AI Chair*

Reviewed on OpenReview: *https://openreview.net/forum?id=wC4ZID0H9a*

## Abstract

In model-based optimisation (MBO) we are interested in using machine learning to design candidates that maximise some measure of reward with respect to a black box function called the (ground truth) oracle, which is expensive to compute since it involves executing a real world process. In offline MBO we wish to do so without assuming access to such an oracle during training or validation, with makes evaluation non-straightforward. While an approximation to the ground oracle can be trained and used in place of it during model validation to measure the mean reward over generated candidates, the evaluation is approximate and vulnerable to adversarial examples. Measuring the mean reward of generated candidates over this approximation is one such 'validation metric', whereas we are interested in a more fundamental question which is finding which validation metrics correlate the most with the ground truth. This involves proposing validation metrics and quantifying them over many datasets for which the ground truth is known, for instance simulated environments. This is encapsulated under our proposed evaluation framework which is also designed to measure extrapolation, which is the ultimate goal behind leveraging generative models for MBO. While our evaluation framework is model agnostic we specifically evaluate denoising diffusion models due to their state-of-the-art performance, as well as derive interesting insights such as ranking the most effective validation metrics as well as discussing important hyperparameters.

## 1 Introduction

In model-based optimisation (MBO), we wish to learn a model of some unknown objective function $f : \mathcal{X} \to \mathcal{Y}$ where $f$ is the ground truth 'oracle', $\boldsymbol{x} \in \mathcal{X}$ is some characterisation of an input and $y \in \mathbb{R}^+$ is the *reward*. The larger the reward is, the more desirable $\boldsymbol{x}$ is. In practice, such a function (a real world process) is often prohibitively expensive to compute because it involves executing a real-world process. For instance if $\boldsymbol{x} \in \mathcal{X}$ is a specification of a protein and the reward function is its potency regarding a particular target in a cell, then synthesising and testing the protein amounts to laborious work in a wet lab. In other cases, synthesising and testing a candidate may also be dangerous, for instance components for vehicles or aircraft. In MBO, we want to learn models that can *extrapolate* – that is, generate inputs whose rewards are beyond that of what we have seen in our dataset. However, we also need ot be rigorous in how we evaluate our models since generating the wrong designs can come at a time, safety, or financial cost.

---

† Work done while author was interning at ServiceNow Research.

Model evaluation in MBO is not straightforward. Firstly, it does not adhere to a typical train/valid/test pipeline that one would expect in other problems. Secondly, it involves evaluating samples that are not from the same distribution as the training set (after all, we want to extrapolate beyond the training set). To understand these difficulties more precisely, we give a quick refresher on a typical training and evaluation pipeline for generative models. If we assume the setting of empirical risk minimisation (Vapnik, 1991) then the goal is to find parameters $\theta^*$ which minimise some training loss $\ell$ on the training set $\mathcal{D}_{\text{train}}$:

$$\theta^* = \arg\min_{\theta} \ \mathcal{L}(\mathcal{D}_{\text{train}}; \theta) = \mathbb{E}_{\boldsymbol{x}, y \sim \mathcal{D}_{\text{train}}} \ell(\boldsymbol{x}, y; \theta), \tag{1}$$

for any model of interest that is parameterised by $\theta$, e.g. a generative model $p_\theta(\boldsymbol{x})$. Since we do not wish to overfit the training set, model selection is performed on a validation set $\mathcal{D}_{\text{valid}}$ and we can write a variant of Equation 1 where the actual model we wish to retain is the following:

$$\theta^* = \arg\min_{\theta \in \Theta} \mathcal{M}(\mathcal{D}_{\text{valid}}; \theta) = \mathbb{E}_{\boldsymbol{x}, y \sim \mathcal{D}_{\text{valid}}} m(\boldsymbol{x}, y; \theta), \tag{2}$$

where $\mathcal{M}$ is a validation metric and $\Theta = \{\theta_j^*\}_{j=1}^m$ comprises a collection of models $\theta_j$, i.e. each of them is the result of optimising Equation 1 under a different hyperparameter configuration or seed. Whatever the best model is according to Equation 2 is finally evaluated on the test set $\mathcal{D}_{\text{test}}$ as an unbiased measure of performance. Since $\mathcal{D}_{\text{test}}$ already comes with labels from the ground truth, it simply suffices to just evaluate Equation 2 and report the result. However, in MBO we wish to *generate* new examples (in particular ones with high reward), which is akin to generating our own 'synthetic' test set. However, we don't know the true values of $y$ (the reward) of its examples unless we evaluate the ground truth oracle $f$ on each generated example, which is very expensive. Secondly, the synthetic test set that we have generated is not intended to be from the same distribution as the training set, since the goal of MBO is to extrapolate and generate examples conditioned on larger rewards than what was observed in the original dataset. This means that the validation set in Equation 2 should *not* be assumed to be the same distribution as the training set, and evaluation should reflect this nuance.

To address the first issue, the most reliable thing to do is to simply evaluate the ground truth on our synthetic test set, but this is extremely expensive since each 'evaluation' of the ground truth $f$ means executing a real world process (i.e. synthesising a protein). Furthermore, since we focus on offline MBO we cannot assume an active learning setting during training like Bayesian optimisation where we can construct a feedback loop between the oracle and the training algorithm. Alternatively, we could simply substitute the ground truth with an approximate oracle $\tilde{f}(\boldsymbol{x})$, but it isn't clear how reliable this is due to the issue of adversarial examples (see Paragraph 2, Section 1). For instance, $\tilde{f}$ could overscore examples in our generated test set and lead us to believe our generated exmples are much better than what they actually are.

Although this is an unavoidable issue in offline MBO, we can still try to alleviate it by finding validation metrics (i.e. $\mathcal{M}$ in Equation 2) which correlate well with the ground truth over a range of datasets where it is known and cheap to evaluate, for instance simulated environments. If we can do this over such datasets then we can those empirical findings to help determine what validation metric we should use for a real world offline problem, where the ground truth is not easily accessible (see Figure 1). To address the second issue, we intentionally construct our train/valid/test pipeline such that the validation set is designed to contain examples with larger reward than the training set, and this sets our work apart from existing literature examining generative models in MBO.

We note that this is very similar to empirical research in reinforcement learning, where agents are evaluated under simulated environments for which the reward function is known and can be computed in silico. However, these environments are ultimately there to inform a much grander goal, which is to have those agents operate safely and reliably in the real world under real world reward functions.

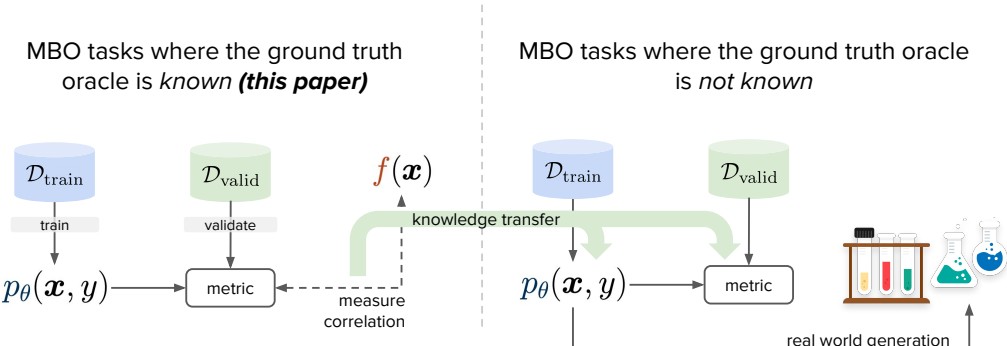

Figure 1: We want to produce designs $\boldsymbol{x}$ that have high reward according to the ground truth oracle $y = f(\boldsymbol{x})$, but this is usually prohibitively expensive to compute since it involves executing a real-world process. If we instead considered datasets where the ground truth oracle is cheap to compute (for instance simulations), we can search for cheap-to-compute validation metrics that correlate well with the ground truth. In principle, this can facilitate faster and more economical generation of novel designs for real-world tasks where the ground truth oracle is expensive to compute.

## 1.1 Contributions

Based on these issues we propose a training and evaluation framework which is amenable to finding good validation metrics that correlate well with the ground truth. In addition, we also assume that the training and validation sets are *not* from the same ground truth distribution. We lay out our contributions as follows:[1]

- We propose a conceptual evaluation framework for *generative models* in offline MBO, where we would like to find validation metrics that correlate well with the ground truth oracle. We assume these validation metrics are cheap to compute. While computing said correlations requires the use of datasets where the ground truth oracle can be evaluated cheaply (i.e. simulations), they can still be useful to inform more real world MBO tasks where the ground truth is expensive to evaluate. In that case, finding good validation metrics can potentially provide large economic savings.

- While our proposed evaluation framework is agnostic to the class of generative model, we specifically demonstrate it using the recently-proposed class of denoising diffusion probabilistic models (DDPMs) (Ho et al., 2020). For this class of model, we examine two conditional variants: classifier-based (Dhariwal & Nichol, 2021) and classifier-free (Ho & Salimans, 2022) guidance. Since DDPMs appear to be relatively unexplored in MBO, we consider our empirical results on these class of models to be an *orthogonal contribution* of our work.

- We explore five validation metrics in our work against four datasets in the Design Bench (Trabucco et al., 2022) framework, motivating their use as well as describing their advantages and disadvantages. We run a large scale study over different hyperparameters and rank these validation metrics by their correlation with the ground truth.

- Lastly, we derive some additional insights such as which hyperparameters are most important to tune. For instance, we found that the classifier guidance term is extremely important, which appears to underscore the trade-off between sample quality and sample diversity, which is a commonly discussed dilemma in generative modelling. We also contribute some thoughts on how diffusion models in offline MBO can be bridged with online MBO.

---

[1]Corresponding code can be found here: **https://github.com/christopher-beckham/validation-metrics-offline-mbo**

## 2 Motivation and proposed framework

We specifically consider the subfield of MBO that is *data-driven* (leverages machine learning) and is *offline*. Unlike online, the offline case doesn't assume an active learning loop where the ground truth can periodically be queried for more labels. Since we only deal with this particular instantiation of MBO, for the remainder of this paper we will simply say *MBO* instead of *offline data-driven MBO*. In MBO, very simple approach to generation is to approximate the ground truth oracle $f$ by training an approximate oracle $f_\theta(\boldsymbol{x})$ – a regression model – from some dataset $\mathcal{D}$, and exploiting it through gradient ascent to generate a high reward candidates:

$$\boldsymbol{x}^* = \arg\max_{\boldsymbol{x}} f(\boldsymbol{x}) \approx \arg\max_{\boldsymbol{x}} f_\theta(\boldsymbol{x}), \tag{3}$$

which can be approximated by iteratively running gradient ascent on the learned regression model for $t \in \{1, \ldots, T\}$:

$$\boldsymbol{x}_{t+1} \leftarrow \boldsymbol{x}_t + \eta \nabla_{\boldsymbol{x}} f_\theta(\boldsymbol{x}), \tag{4}$$

and $\boldsymbol{x}_0$ is sampled from some prior distribution. The issue here however is that for most problems, this will produce an input that is either *invalid* (e.g. not possible to synthesise) or is poor yet receives a large reward from the approximate oracle (*overestimation*). This is the case when the space of valid inputs lies on a low-dimensional manifold in a much higher dimension space (Kumar & Levine, 2020). How these problems are mitigated depends on whether one approaches MBO from a discriminative modelling point of view (Fu & Levine, 2021; Trabucco et al., 2021; Chen et al., 2022a), or a generative modelling one (Brookes et al., 2019; Fannjiang & Listgarten, 2020; Kumar & Levine, 2020). For instance, Equation 3 implies a discriminative approach where $f_\theta(\boldsymbol{x})$ is a regression model. However, in Equation 4 it is reinterpreted as an energy model Welling & Teh (2011). While this complicates the distinction between discriminative and generative, we refer to the generative approach as one where it is clear that a joint distribution $p_\theta(\boldsymbol{x}, y)$ is being learned. For instance, if we assume that $p_\theta(\boldsymbol{x}, y) = p_\theta(y|\boldsymbol{x})p_\theta(\boldsymbol{x})$, then the former is a probabilistic form of the regression model and the latter models the likelihood, and modelling some notion of it will almost certainly mitigate the adversarial example issue since it is modelling the prior probability of observing such an input. Because of this, we argue that a generative view of MBO is more appropriate and we will use its associated statistical language for the remainder of this paper.

Let us assume we have trained a conditional generative model of the form $p_\theta(\boldsymbol{x}|y)$ on our training set $\mathcal{D}_{\text{train}}$. If $p_{\text{train}}(y)$ denotes the empirical distribution over the $y$'s in the training set, then this can be used to write the joint distribution as $p_\theta(\boldsymbol{x}, y) = p_\theta(\boldsymbol{x}|y)p_{\text{train}}(y)$. We do not wish to sample from this joint distribution however, because we ultimately want to generate $\boldsymbol{x}$'s whose $y$'s are as large as possible. To do this we could switch out the prior $p_{\text{train}}(y)$ for one that reflects the range of values we wish to sample from. However, it wouldn't be clear if the model has generalised in this regime of $y$'s; for instance the sampled $\boldsymbol{x}$'s may be invalid. What we want to do is be able to measure and select for models (i.e. $p_\theta(\boldsymbol{x}|y)$ for some good $\theta$) that are able to *extrapolate*; in other words, models which assign small loss to examples that *have larger $y$'s than those observed in the training set*. As an example, one such appropriate loss could be the negative log likelihood.

We can measure this through careful construction of our training and validation sets without having to leave the offline setting. Assume the full dataset $\mathcal{D} = \{(\boldsymbol{x}_i, y_i\}_{i=1}^n$ and $(\boldsymbol{x}, y) \sim p(\boldsymbol{x}, y)$, the ground truth joint distribution. Given some threshold $\gamma$, we can imagine dealing with two truncated forms of the ground truth $p_{0,\gamma}(\boldsymbol{x}, y)$ and $p_\gamma(\boldsymbol{x}, y)$, where:

$$p_{0,\gamma}(\boldsymbol{x}, y) = \{(\boldsymbol{x}, y) \sim p(\boldsymbol{x}, y)|y \in [0, \gamma]\}$$
$$p_\gamma(\boldsymbol{x}, y) = \{(\boldsymbol{x}, y) \sim p(\boldsymbol{x}, y)|y \in (\gamma, \infty]\}, \tag{5}$$

where $p_\gamma$ is the distribution of samples which are not seen during training but nonetheless we would like our generative model to explain well. Therefore, if we split $\mathcal{D}$ based on $\gamma$ then we can think of the left split $\mathcal{D}_{\text{train}}$ as a finite collection of samples from $p_\gamma(\boldsymbol{x}, y)$ and the right split $\mathcal{D}_{\text{valid}}$ from $p_{0,\gamma}(\boldsymbol{x}, y)$. We would like to train models on $\mathcal{D}_{\text{train}}$ and maximise some measure of desirability on $\mathcal{D}_{\text{valid}}$ (via hyperparameter tuning) to find the best model.

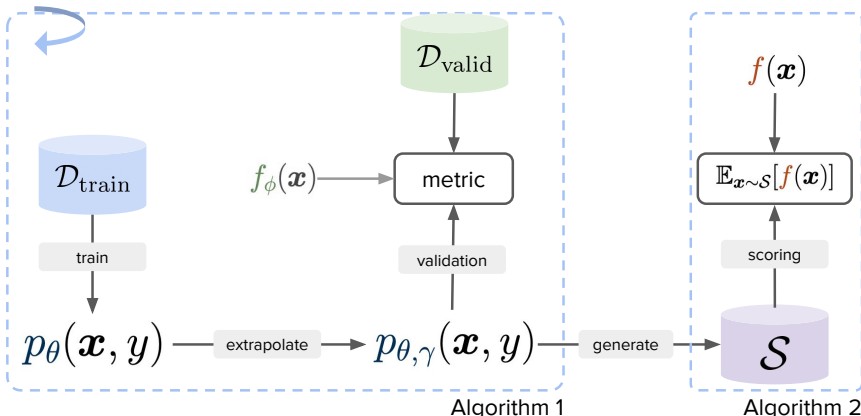

Figure 2: A visualisation of our evaluation framework. Here, we assume joint generative models of the form $p_\theta(\boldsymbol{x}, y)$. Models are trained on $\mathcal{D}_{\text{train}}$ as per Section 2.1, and in this paper we assume the use of conditional denoising diffusion probabilistic models (DDPMs). For this class of model the joint distribution $p_\theta(\boldsymbol{x}, y)$ decomposes into $p_\theta(\boldsymbol{x}|y)p(y)$, and the way we condition the model on $y$ is described in Section 2.1.1. In order to generate samples conditioned on rewards $y$ larger than what was observed in the training set, we must switch the prior distribution of the model, which corresponds to 'extrapolating' it and is described in Section 2.2. Validation is done periodically during training and the best weights are saved for each validation metric considered. The precise details of this are described in Algorithm 1. When the best models have been found we perform a final evaluation on the real ground truth oracle, and this process is described in Algorithm 2.

## 2.1 Training and generation

While our proposed evaluation framework is agnostic to the class of generative model used, in this paper we specifically focus on denoising diffusion probabilistic models (DDPMs) (Sohl-Dickstein et al., 2015; Ho et al., 2020). DDPMs are currently state-of-the-art in many generative tasks and do not suffer from issues exhibited from other classes of model. They can be seen as multi-latent generalisations of variational autoencoders, and just like VAEs they optimise a variational bound on the negative log likelihood of the data[2]. First, let us consider an *unconditional* generative model $p_\theta(\boldsymbol{x})$, whose variational bound is:

$$-\mathbb{E}_{\boldsymbol{x}_0 \sim p(\boldsymbol{x}_0)} \log p_\theta(\boldsymbol{x}_0) \leq \mathbb{E}_{p(\boldsymbol{x}_0,\ldots,\boldsymbol{x}_T)}\left[ \log \frac{p(\boldsymbol{x}_1,\ldots,\boldsymbol{x}_T|\boldsymbol{x}_0)}{p_\theta(\boldsymbol{x}_0,\ldots,\boldsymbol{x}_T)} \right] \tag{6}$$

$$= \mathbb{E}_{p(\boldsymbol{x}_0,\ldots,\boldsymbol{x}_T)}\left[ \underbrace{\log \frac{p(\boldsymbol{x}_T)}{p(\boldsymbol{x}_T|\boldsymbol{x}_0)}}_{L_T} + \sum_{t>1} \underbrace{\log \frac{p_\theta(\boldsymbol{x}_{t-1}|\boldsymbol{x}_t)}{p(\boldsymbol{x}_{t-1}|\boldsymbol{x}_t, \boldsymbol{x}_0)}}_{L_t} - \underbrace{\log p_\theta(\boldsymbol{x}_0|\boldsymbol{x}_1)}_{L_0} \right], \tag{7}$$

Here, $\boldsymbol{x}_0 \sim p(\boldsymbol{x}_0)$ is the real data (we use $\boldsymbol{x}_0$ instead of $\boldsymbol{x}$ to be consistent with DDPM notation, but they are the same), and $p(\boldsymbol{x}_t|\boldsymbol{x}_{t-1})$ for $t \in \{1, \ldots, T\}$ is a predefined Gaussian distribution which samples a noisy $\boldsymbol{x}_t$ which contains more noise than $\boldsymbol{x}_{t-1}$, i.e. as $t$ gets larger the original data $\boldsymbol{x}_0$ progressively becomes more noised. These conditional distributions collectively define the *forward process*. The full joint distribution of this forward process is[3]:

$$p(\boldsymbol{x}_0,\ldots,\boldsymbol{x}_T) = p(\boldsymbol{x}_0) \prod_{t=1}^{T} p(\boldsymbol{x}_t|\boldsymbol{x}_{t-1}), \tag{8}$$

---

[2]Typical DDPM literature uses $q$ to define the real distribution, but here we use $p$ to be consistent with earlier notation, though it is not to be confused with $p_\theta$, the learned distribution.

[3]Note we have still used orange to denote that $p(\boldsymbol{x}_0)$ is the ground truth data distribution, but we have omitted it for $p(\boldsymbol{x}_t|\boldsymbol{x}_{t-1})$ to make it clear that these are predefined noise distributions.

and each conditional noising distribution has the form:

$$p(\boldsymbol{x}_t|\boldsymbol{x}_{t-1}) = \mathcal{N}(\boldsymbol{x}_t; \sqrt{1 - \beta_t}\boldsymbol{x}_{t-1}, \beta_t\mathbf{I}), \tag{9}$$

where $\beta_t$ follows a predefined noise schedule. Since the product of Gaussians are also Gaussian, one can also define the $t$-step noising distribution $p(\boldsymbol{x}_t|\boldsymbol{x}_0)$ which is more computationally efficient:

$$p(\boldsymbol{x}_t|\boldsymbol{x}_0) = \mathcal{N}(\boldsymbol{x}_t; \sqrt{\bar{\alpha}_t}\boldsymbol{x}_0 + (1 - \bar{\alpha}_t)\mathbf{I}) \tag{10}$$

$$\implies \boldsymbol{x}_t = \sqrt{\bar{\alpha}_t}\boldsymbol{x}_0 + \sqrt{(1 - \bar{\alpha}_t)}\epsilon, \quad \epsilon \sim \mathcal{N}(0, \mathbf{I}) \tag{11}$$

where $\alpha_t = 1 - \beta_t$ and $\bar{\alpha}_t = \prod_{s=1}^{t}\alpha_s$. Generally speaking, we wish to learn a neural network $p_\theta(\boldsymbol{x}_{t-1}|\boldsymbol{x}_t)$ to undo noise in the forward process, and these learned conditionals comprise a joint distribution for the *reverse* process:

$$p_\theta(\boldsymbol{x}_{t-1}|\boldsymbol{x}_t) = \mathcal{N}(\boldsymbol{x}_{t-1}; \mu_\theta(\boldsymbol{x}_t, t); \beta_{t-1}\mathbf{I}), \tag{12}$$

where $p_\theta$ is expressed via a neural network $\mu_\theta$ which learns to predict the mean of the conditional distribution for $\boldsymbol{x}_{t-1}$ given $\boldsymbol{x}_t$, and we wish to find parameters $\theta$ to minimise the expected value of Equation 6 over the entire dataset. In practice however, if the conditionals for $p_\theta$ and $p$ are assumed to be Gaussian, one can dramatically simplify Equation 6 and re-write $L_t$ (for any $t$) as a noise prediction task. where instead we parameterise a neural network $\epsilon_\theta(\boldsymbol{x}_t, t)$ to predict the *noise* from $\boldsymbol{x}_t \sim p(\boldsymbol{x}_t|\boldsymbol{x}_0)$, as shown in Equation 11 via the reparamterisation trick. Note that $\epsilon_\theta(\boldsymbol{x}_t, t)$ and $\mu_\theta(\boldsymbol{x}_t, t)$ differ by a closed form expression (and likewise with $p_\theta(\boldsymbol{x}_{t-1}|\boldsymbol{x})$), but for brevity's sake we defer those details to Ho et al. (2020) and simply state the final loss function to be:

$$\mathbb{E}_{\boldsymbol{x}_0 \sim p(\boldsymbol{x}_0), t \sim U(1,T), \epsilon \sim \mathcal{N}(0,\mathbf{I})}\big[\|\epsilon - \epsilon_\theta(\sqrt{\bar{\alpha}_t}\boldsymbol{x}_0 + \sqrt{1 - \bar{\alpha}_t}\epsilon, t)\|^2\big], \tag{13}$$

where $\sqrt{\bar{\alpha}_t}\boldsymbol{x}_0 + \sqrt{1 - \bar{\alpha}_t}\epsilon_t$ is simply just writing out $\boldsymbol{x}_t \sim p(\boldsymbol{x}_t|\boldsymbol{x}_0)$ via the reparamterisation trick (Equation 11), and $\{\alpha_t\}_{t=1}^{T}$ defines a noising schedule. Since we have defined our training set $\mathcal{D}_{\text{train}}$ to be samples from $p_{0,\gamma}(\boldsymbol{x}, y)$, our training loss is simply:

$$\min_\theta \mathcal{L}_{\text{DSM}}(\theta) = \min_\theta \mathbb{E}_{\boldsymbol{x}_0 \sim \mathcal{D}_{\text{train}}, \epsilon_t \sim \mathcal{N}(0,\mathbf{I})}\big[\|\epsilon_t - \epsilon_\theta(\sqrt{\bar{\alpha}_t}\boldsymbol{x}_0 + \sqrt{1 - \bar{\alpha}_t}\epsilon_t, t)\|^2\big]. \tag{14}$$

In order to generate samples, stochastic Langevin dynamics (SGLD) is used in conjunction with the noise predictor $\epsilon_\theta$ to construct a Markov chain that by initialising $\boldsymbol{x}_T \sim p(\boldsymbol{x}_T) = \mathcal{N}(0, \mathbf{I})$ and running the following equation for $t \in \{T-1, \ldots, 0\}$:

$$\boldsymbol{x}_{t-1} = \frac{1}{\sqrt{\alpha_t}}\Big(\boldsymbol{x}_t - \frac{1 - \alpha_t}{\sqrt{1 - \bar{\alpha}_t}}\epsilon_\theta(\boldsymbol{x}_t, t)\Big) + \beta_t\boldsymbol{z}, \quad \boldsymbol{z} \sim \mathcal{N}(0, \mathbf{I}). \tag{15}$$

Note that while the actual neural network that is *trained* is a noise predictor $\epsilon_\theta(\boldsymbol{x}_t, t)$ which in turn has a mathematical relationship with the *reverse conditional* $p_\theta(\boldsymbol{x}_{t-1}|\boldsymbol{x}_t)$ we will generally refer to the diffusion model as simply $p_\theta(\boldsymbol{x})$ throughout the paper, since the use of Equation 15 implies a sample from the distribution $p_\theta(\boldsymbol{x}_0)$, and we have already defined $\boldsymbol{x} = \boldsymbol{x}_0$.

### 2.1.1 Conditioning

**Classifier-based guidance** Note that Equation 14 defines an *unconditional* model $p_\theta(\boldsymbol{x})$. In order to be able to condition on the reward $y$, we can consider two options.[4] The first is *classifier-based guidance* (Dhariwal & Nichol, 2021). Here, we train an *unconditional* diffusion model $p_\theta(\boldsymbol{x})$, but during generation we define a *conditional* noise predictor which leverages pre-trained classifier $p_\theta(y|\boldsymbol{x}_t)$ which predicts the reward from $\boldsymbol{x}_t$:

$$\epsilon_\theta(\boldsymbol{x}_t, t, y) = \underbrace{\epsilon_\theta(\boldsymbol{x}_t, t) - \sqrt{1 - \bar{\alpha}_t}w\nabla_{\boldsymbol{x}_t}\log p_\theta(y|\boldsymbol{x}_t; t)}_{\text{classifier-based guidance}} \tag{16}$$

---

[4]While it is possible to derive a conditional ELBO from which a DDPM can be derived from, the most popular method for conditioning in DDPMs appears to be via guiding an unconditional model instead.

where $p_\theta(y|\boldsymbol{x})$ is also trained on $\mathcal{D}_{\text{train}}$ and $w \in \mathbb{R}^+$ is a hyperparameter which balances sample diversity and sample quality. Equation 16 is then plugged into the SGLD algorithm (Equation 15) to produce a sample from the conditional distribution $p_\theta(\boldsymbol{x}|y)$. Note that since $p_\theta(y|\boldsymbol{x}_t;t)$ is meant to condition on $\boldsymbol{x}_t$ for any $t$, it ideally requires a similar training setup to that of the diffusion model, i.e. sample different $\boldsymbol{x}_t$'s via Equation 10 train the network to predict $y$.

**Classifier-free guidance**  In *classifier-free guidance* (Ho & Salimans, 2022), the noise predictor is re-parameterised to support conditioning on label $y$, but it is stochastically 'dropped' during training with some probability $\tau$, in which case $y$ is replaced with a null token:

$$\mathcal{L}_{\text{C-DSM}}(\theta;\tau) = \mathbb{E}_{(\boldsymbol{x}_0,y) \sim \mathcal{D}_{\text{train}}, t \sim U(1,T), \lambda \sim U(0,1)} \big[ \|\epsilon - \epsilon_\theta(\sqrt{\bar{\alpha}_t}\boldsymbol{x}_0 + \sqrt{1-\bar{\alpha}_t}\epsilon, \mathbf{1}_{\lambda<\tau}(y), t)\|^2 \big]. \tag{17}$$

where $\mathbf{1}_{\lambda<\tau}(y)$ is the indicator function and returns a null token if $\lambda < \tau$ otherwise $y$ is returned. The additional significance of this is that at generation time, one can choose various tradeoffs of (conditional) sample quality and diversity by using the following noise predictor, for some hyperparameter $w$:

$$\bar{\epsilon}_\theta(\boldsymbol{x}_t,t,y) = \underbrace{(w+1)\epsilon_\theta(\boldsymbol{x}_t,t,y) - w\epsilon_\theta(\boldsymbol{x},t)}_{\text{classifier-free guidance}} \tag{18}$$

## 2.2   Extrapolation

Given one of the two conditioning variants in Section 2.1.1, we can denote our generative model as $p_\theta(\boldsymbol{x}|y)$, which has been trained on $(\boldsymbol{x},y)$ pairs from $\mathcal{D}_{\text{train}}$. If we denote the distribution over $y$'s in the training set as $p_{0,\gamma}(y)$ then through Bayes' rule we can write the joint likelihood as: $p_\theta(\boldsymbol{x},y) = p_\theta(\boldsymbol{x}|y)p_{0,\gamma}(y)$. This equation essentially says: to generate a sample $(\boldsymbol{x},y)$ from the joint distribution defined by the model, we first sample $y \sim p_{0,\gamma}(y)$, then we sample $\boldsymbol{x} \sim p_\theta(\boldsymbol{x}|y)$ via SGLD (Equation 15). Samples from $p_{0,\gamma}(y)$ can be approximated by sampling from the empirical distribution of $y$'s over the training set. Decomposing the joint distribution into a likelihood and a prior over $y$ means that we can change the latter at any time. For instance, if we wanted to construct an 'extrapolated' version of this model, we can simply replace the prior in this equation with $p_\gamma(y)$, which is the prior distribution over $y$ for the validation set. We define this as the *extrapolated model* (Figure 2, *extrapolate* caption):[5]

$$p_{\theta,\gamma}(\boldsymbol{x},y) = p_\theta(\boldsymbol{x}|y)p_\gamma(y) \tag{19}$$

and samples $p_\gamma(y)$ can be approximated by sampling from the empirical distribution over $y$ from the validation set. Of course, it is not clear to what extent the extrapolated model would be able to generate high quality inputs from $y$'s much larger than what it has observed during training. This is why we need to perform model selection via the use of some validation metric that characterises the model's ability to extrapolate.

## 2.3   Model selection

Suppose we trained multiple diffusion models, each model differing by their set of hyperparameters. If we denote the $j$'th model's weights as $\theta_j$, then model selection amounts to selecting a $\theta^* \in \Theta = \{\theta_j\}_{j=1}^m$ which minimises some 'goodness of fit' measure on the validation set. We call this a *validation metric*, which is computed the held-out validation set $\mathcal{D}_{\text{valid}}$. One such metric that is fit for a generative model would be the expected log likelihood over samples in the validation set, assuming it is tractable:

$$\theta^* = \arg\max_{\theta \in \Theta} \frac{1}{|\mathcal{D}_{\text{valid}}|} \sum_{(\boldsymbol{x},y) \in \mathcal{D}_{\text{valid}}} \log p_{\theta,\gamma}(\boldsymbol{x},y) \tag{20}$$

where $p_{\theta,\gamma}(\boldsymbol{x},y) = p_\theta(\boldsymbol{x}|y)p_\gamma(y)$ is the extrapolated model as originally described in Equation 19. Since we have already established that the validation set comprises candidates that are higher scoring than the training set, this can be thought of as selecting for models which are able to *explain* (i.e. assign high conditional

---

[5]There are other ways to infer an extrapolated model from a 'base' model, and we describe some of these approaches in Section 3.

likelihood) to those samples. However, we cannot compute the conditional likelihood since it isn't tractable, and nor can we use its evidence lower bound (ELBO). This is because both conditional diffusion variants (Section 2.1.1) assume a model derived from the unconditional ELBO which is a bound on $p_\theta(\boldsymbol{x})$, rather than the conditional ELBO is correspondingly a bound on $p_\theta(\boldsymbol{x}|y)$. Despite this, if we are training the classifier-free diffusion variant, we can simply use Equation 17 as a proxy for it but computed over the validation set:

$$\mathcal{M}_{\text{C-DSM}}(\mathcal{D}_{\text{valid}}; \theta) = \frac{1}{\mathcal{D}_{\text{valid}}} \sum_{(\boldsymbol{x}_0, y) \sim \mathcal{D}_{\text{valid}}} \|\epsilon_t - \epsilon_\theta(\boldsymbol{x}_t, y, t)\|^2, \tag{21}$$

where $\boldsymbol{x}_t \sim p(\boldsymbol{x}_t|\boldsymbol{x}_0)$. Equation 21 can be thought of as selecting for models which are able to perform a good job of conditionally *denoising* examples from the validation set. While Equation 21 seems reasonable, arguably it is not directly measuring what we truly want, which is a model that can generate candidates that are high rewarding as possible. Therefore, we should also devise a validation metric which favours models that, when 'extrapolated' during generation time (i.e. we sample $\boldsymbol{x}|y$ where $y$'s are drawn from $p_\gamma(y)$), are likely under an approximation of the ground truth oracle. This approximate oracle is called the *validation oracle $f_\phi$*, and is trained on both $\mathcal{D}_{\text{train}} \cup \mathcal{D}_{\text{valid}}$ (see Figure 2):

$$\theta^* = \arg\max_{\theta \in \Theta} \mathbb{E}_{\tilde{\boldsymbol{x}}, y \sim p_{\theta, \gamma}(\boldsymbol{x}, y)} f_\phi(\tilde{\boldsymbol{x}}) = \arg\max_{\theta \in \Theta} \mathbb{E}_{\tilde{\boldsymbol{x}} \sim p_\theta(\boldsymbol{x}|y), y \sim p_\gamma(y)} f_\phi(\tilde{\boldsymbol{x}}). \tag{22}$$

We consider a biased variant of Equation 22 where the expectation is computed over the *best* 128 samples generated by the model, to be consistent with Trabucco et al. (2022). This can be written as the following equation:

$$\mathcal{M}_{\text{reward}}(\mathcal{S}; \theta, \phi) = \frac{1}{K} \sum_{i=1}^{K} f_\phi\big(\text{sorted}(\mathcal{S}; f_\phi)_i\big), \quad \mathcal{S}_i \sim p_{\theta, \gamma}(\boldsymbol{x}, y) \tag{23}$$

where $\text{sorted}(\mathcal{S}, f_\phi)$ sorts $\mathcal{S} = \{\tilde{\boldsymbol{x}}_i\}_i$ in descending order via the magnitude of prediction, and $|\mathcal{S}| \gg K$.

Lastly, there are two additional validation metrics we propose, and these have been commonly used for adversarial-based generative models (which do not permit likelihood evaluation). For the sake of space we defer the reader to Section A.1 and simply summarise them below:

- Fréchet distance (Heusel et al., 2017) (Equation S28): given samples from the validation set and samples from the extrapolated model, fit multivariate Gaussians to both their embeddings (a hidden layer in $f_\phi$ is used for this) and measure the distance between them. This can be thought of as a likelihood-free way to measure the distance between two distributions. We denote this as $\mathcal{M}_{\text{FD}}$.

- Density and coverage (Equation S32): Naeem et al. (2020) proposed 'density' and 'coverage' as improved versions of precision and recall, respectively (O'Donoghue et al., 2020; Kynkäänniemi et al., 2019). In the generative modelling literature, precision and recall measure both sample quality and mode coverage, i.e. the extent to which the generative model can explain samples from the data distribution. We denote this as $\mathcal{M}_{\text{DC}}$, which is a simple sum over the density and coverage metric.

We summarise all validation metrics – as well as detail their pros and cons – in Table 1. We also summarise all preceding subsections in Algorithm 1, which constitutes our evaluation framework.

### 2.3.1 Final evaluation

We may now finally address the core question presented in this paper: what validation metrics are best correlated with the ground truth? Since we have already defined various validation metrics in Section 2.3, the only thing left is to define an additional metric – a *test metric* – which is a function of the ground truth oracle. We simply define this to be an unbiased estimate of Equation 23 which uses the ground truth oracle $f$ to rank the top 128 candidates:

$$\mathcal{M}_{\text{test-reward}}(\mathcal{S}; \theta, \phi) = \frac{1}{K} \sum_{i=1}^{K} f\big(\text{sorted}(\mathcal{S}; f_\phi)_i\big), \quad \mathcal{S}_i \sim p_{\theta, \gamma}(\boldsymbol{x}, y) \tag{24}$$

---

**Algorithm 1** Training algorithm, with early implicit early stopping. For $m$ predefined validation metrics, the best weights for each along with their values and stored and returned. (For consistency of notation, each validation metric is of the form $\mathcal{M}_j(\boldsymbol{X}, \tilde{\boldsymbol{X}}, \theta, \phi)$, though the true arguments that are taken may be a subset of these. See Table 1.)

---

**Require:**
  Threshold $\gamma$, s.t. $\mathcal{D}_{\text{train}} \sim p_{0,\gamma}(\boldsymbol{x}, y), \mathcal{D}_{\text{valid}} \sim p_\gamma(\boldsymbol{x}, y)$                    ▷ Eqns. 5
  Number of training epochs $n_{\text{epochs}}$, validation rate $n_{\text{eval}}$
  Validation metrics (functions): $\{\mathcal{M}_j\}_{j=1}^m$, $\mathcal{M}_j$ has arguments $\mathcal{M}_j(\mathcal{D}, \tilde{\mathcal{D}}, \theta, \phi)$
  $h \in \mathcal{H}$ hyperparameters used to initialise $\theta$
  Validation oracle $f_\phi(\boldsymbol{x})$                    ▷ pre-trained on train + val $\mathcal{D}_{\text{train}} \cup \mathcal{D}_{\text{valid}}$ set
 1: $\boldsymbol{X}_{\text{train}}, \boldsymbol{Y}_{\text{train}} \leftarrow \mathcal{D}_{\text{train}}, \boldsymbol{X}_{\text{valid}}, \boldsymbol{Y}_{\text{valid}} \leftarrow \mathcal{D}_{\text{valid}}$
 2: $\theta \leftarrow \text{initialise}(h)$
 3: $\text{best\_weights} \leftarrow \{\theta\}_{j=1}^m$                    ▷ store best weights so far for each validation metric
 4: $\text{best\_metrics} \leftarrow \{\infty\}_{j=1}^m$                    ▷ store best (smallest) values seen per metric
 5: **for** epoch in $\{1, \dots, n_{\text{epochs}}\}$ **do**
 6:     sample $(\boldsymbol{x}, y) \sim \mathcal{D}_{\text{train}}$
 7:     $\theta \leftarrow \theta - \eta \nabla_\theta \mathcal{L}_\theta(\boldsymbol{x}, y)$                    ▷ $\mathcal{L}$: eqn. 14, with either eqn. 16 or eqn. 18
 8:     **if** epoch % $n_{\text{eval}} = 0$ **then**
 9:         ▷ **Model selection (Sec. 2.3)**
10:         $\tilde{\mathcal{D}} \leftarrow \{(\tilde{\boldsymbol{x}}_i, y_i)\}_{i=1}^{|\mathcal{D}_{\text{valid}}|}$, where $\tilde{\boldsymbol{x}}_i \sim p_\theta(\boldsymbol{x}|y_i)$ and $y_i = (\boldsymbol{Y}_{\text{valid}})_i$     ▷ sample using eqn. 15
11:         ▷ **Evaluate validation metrics (Table 1)**
12:         **for** $j$ in $\{1, \dots, m\}$ **do**
13:             $m_j \leftarrow \mathcal{M}_j(\mathcal{D}_{\text{valid}}, \tilde{\mathcal{D}}, \theta, \phi)$
14:             **if** $m_j < \text{best\_metrics}_j$ **then**
15:                 $\text{best\_metrics}_j \leftarrow m_j$                    ▷ found new best metric value for metric $j$
16:                 $\text{best\_weights}_j \leftarrow \theta$                    ▷ save new weights for metric $j$
17:             **end if**
18:         **end for**
19:     **end if**
20: **end for**
21: **return** best\_weights, best\_metrics

---

**Algorithm 2** Final evaluation algorithm. As per Algorithm 1, each validation metric $\mathcal{M}_j$ is associated with the best weights $\theta_j$ found through hyperparameter tuning and early stopping. For each $\theta_j$ we generate a candidate set $\mathcal{S}_i$ of examples, retain the $K$ best candidates as per the predicted reward from validation oracle, and then compute the real reward on the ground truth oracle.

---

**Require:**
  best\_weights $= \{\theta_j\}$                    ▷ assumed to be the best weights found for each validation metric $\mathcal{M}_j$
 1: $\text{test\_rewards} \leftarrow \{\}_{j=1}^m$
 2: **for** $j$ in $\{1, \dots, m\}$ **do**
 3:     $\theta \leftarrow \text{best\_weights}_j$                    ▷ Load in best weights for metric $j$
 4:     $\mathcal{S} \leftarrow \{\tilde{\boldsymbol{x}}_i\}_{i=1}^N$, where $\tilde{\boldsymbol{x}}_i \sim p_\theta$ and $y_i \sim \boldsymbol{Y}_{\text{valid}}$
 5:     $\text{valid\_rewards} = \{f_\phi(\mathcal{S}_i)\}_{i=1}^N$                    ▷ predict reward for each generated candidate
 6:     $\pi = \text{argsort}(\text{valid\_scores})_{1,\dots,K}$                    ▷ get top $K$ examples wrt to $f_\phi$ predictions
 7:     $\text{test\_rewards}_j \leftarrow \{f(\mathcal{S}_{\pi(i)})\}_{i=1}^K$                    ▷ get unbiased estimate of test rewards
 8: **end for**
 9: **return** test\_rewards

---

This is the test metric which is used to determine how well correlated our validation metrics are with the ground truth. For instance, suppose we have trained many different generative models via Algorithm 1 – each model corresponding to a different set of hyperparameters – if we run Algorithm 2 on each of these

models then we can compute quantitative metrics such as correlation between **best_metrics** (Alg. 1) and **test_rewards** (Alg. 2), and indeed this is what we will be demonstrating in Section 4.

While it seems reasonable to assume that Equation 24 would be most correlated with its validation set equivalent (Equation 23), this would only be the case in the limit of a sufficient number of examples to be used to train the validation oracle $f_\phi$. Otherwise, the less data that is used the more it will be prone to overscoring 'adversarial' examples produced by the generative model. Therefore, it may be wise to consider validation metrics which put less emphasis on the magnitude of predictions produced by the oracle, e.g. measuring the distance between distributions.

Table 1: The column 'distance?' asks whether the validation metric is measuring some form of distance between the extrapolated distribution $p_\gamma(\boldsymbol{x}, y)$ and the extrapolated model $p_{\theta,\gamma}(\boldsymbol{x}, y)$. †: RKL = reverse KL divergence, see Sec. A.2.2 for proof it is an approximation of this divergence; ‡: FKL = forward KL divergence, since diffusion models optimise an evidence lower bound (Eqn. 6); ∗ = distances here are measured not in data space, but in the semantic space defined by one of the hidden layers inside $f_\phi$.

| | eqn. | require $f_\phi$ | model agnostic? | distance? |
|---|---|---|---|---|
| $\mathcal{M}_{\text{C-DSM}}(\mathcal{D}_{\text{valid}}; \theta)$ | 21 | ✗ | ✗ | (approx. FKL)‡ ✓ |
| $-\mathcal{M}_{\text{reward}}(\mathcal{S}; \theta, \phi)$ | 23 | ✓ | ✓ | ✗ |
| $\mathcal{M}_{\text{Agr}}(\mathcal{D}_{\text{valid}}; \theta, \phi)$ | 26 | ✓ | ✓ | (approx. RKL)† ✓ |
| $\mathcal{M}_{\text{FD}}(\boldsymbol{X}_{\text{valid}}, \tilde{\boldsymbol{X}}; \phi)$ | S28 | ✓ | ✓ | Fréchet∗ ✓ |
| $-\mathcal{M}_{\text{DC}}(\boldsymbol{X}_{\text{valid}}, \tilde{\boldsymbol{X}}; \phi)$ | S32 | ✓ | ✓ | ∗✓ |

## 3   Related work

**Design Bench**   Design Bench is an evaluation framework by Trabucco et al. (2022) that facilitates the training and evaluation of MBO algorithms. Design Bench, as of time of writing, provides *four discrete* and *four continuous* datasets. These datasets can be further categorised based on two attributes: whether a ground truth oracle exists or not, and whether the input distribution is fully observed (i.e. the combined train and test splits contain all possible input combinations). In terms of evaluation, Design Bench does not prescribe a validation set (only a training set and test oracle), which we argue is important in order to address the core question of our work, which is finding validation metrics that correlate well with the ground truth oracle. While Trabucco et al. (2022) does allude to validation sets in the appendix, these do not convey the same semantic meaning as our validation set since theirs is assumed to be a subsample of the training set, and therefore come from the training distibution. Lastly, while the same appendix provides examples for different validation metrics per baseline, the overall paper itself is concerned with establishing reliable benchmarks for comparison, rather than comparing validation metrics directly.

**Validation metrics**   To be best of our knowledge, a rigorous exploration of validation metrics has not yet been explored in MBO. The choice of validation metric is indeed partly influenced by the generative model, since one can simply assign the validation metric to be the same as the training loss but evaluated on the validation set. For example, if we choose likelihood-based generative models (essentially almost all generative models apart from GANs), then we can simply evaluate the likelihood on the validation set and use that as the validation metric (Equation 20). However, it has been well established that likelihood is a relatively poor measure of sample quality and is more biased towards sample coverage (Huszár, 2015; Theis et al., 2015; Dosovitskiy & Brox, 2016). While GANs have made it difficult to evaluate likelihoods – they are non-likelihood-based generative models – it has fortunately given rise to an extensive literature proposing 'likelihood-free' evaluation metrics (Borji, 2022), and these are extremely useful to explore for this study for two reasons. Firstly, likelihood-free metrics are *agnostic* to the class of generative model used, and secondly they are able to probe various aspects of generative models that are hard to capture with just likelihood. As an example, the Fréchet Distance (FID) (Heusel et al., 2017) is commonly used to evaluate the realism of generated samples with respect to a reference distribution, and correlates well with human judgement of sample quality. Furthermore, metrics based on precision and recall can be used to quantify sample quality and sample coverage, respectively (Sajjadi et al., 2018; Kynkäänniemi et al., 2019).

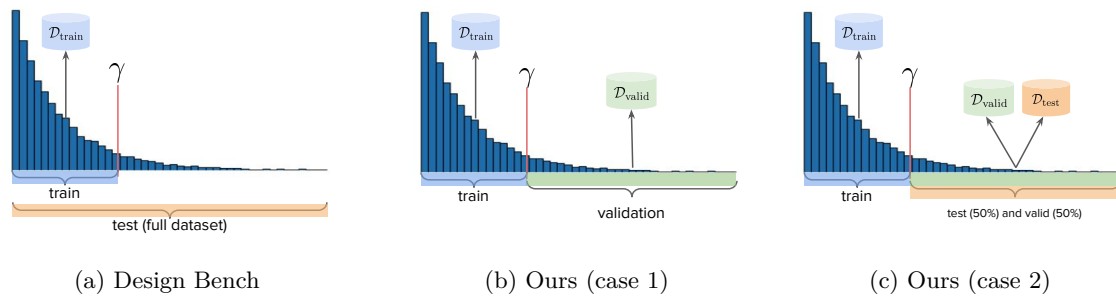

(a) Design Bench       (b) Ours (case 1)       (c) Ours (case 2)

Figure 3: 3a: Design Bench only prescribes a training split which is determined by a threshold $\gamma$ to only filter examples whose $y$'s are less than or equal to this threshold. The full dataset, while technically accessible, is not meant to be accessed for model selection as per the intended use of the framework. While the training set could be subsampled to give an 'inner' training set and validation set, the validation set would still come from the same distribution as training, which means we cannot effectively measure how well a generative model extrapolates. To address this, we retain the training set but denote everything else (examples whose rewards are $> \gamma$) to be the validation set (3b), and the validation oracle $f_\phi$ is trained on $\mathcal{D}_{\text{train}} \cup \mathcal{D}_{\text{valid}}$. No test set needs to be created since the ground truth oracle $f$ is the 'test set'. However, if the ground truth oracle does not exist because the MBO dataset is not exact, we need to also prescribe a test set (3c). Since there is no ground truth oracle $f$, we must train a 'test oracle' $\tilde{f}$ on $\mathcal{D}_{\text{train}} \cup \mathcal{D}_{\text{valid}} \cup \mathcal{D}_{\text{test}}$ (i.e. the full dataset). Note that this remains compatible with the test oracles prescribed by Design Bench, since they are also trained on the full data. Furthermore, our training sets remain identical to theirs.

**Use of validation set** Compared to other works, the use of a validation set varies and sometimes details surrounding how the data is split is opaque. For example, in Kumar & Levine (2020) there is no mention of a training or validation set; rather, we assume that only $\mathcal{D}_{\text{train}}$ and $\mathcal{D}_{\text{test}}$ exists, with the generative model being trained on the former and test oracle on the latter (note that if the test oracle is approximate there is no need for a $\mathcal{D}_{\text{test}}$). This also appears to be the case for Fannjiang & Listgarten (2020). While Design Bench was proposed to standardise evaluation, its API does not prescribe a validation set[6]. While the training set could in principle be subsetted into a smaller training set and a validation set (such as in Qi et al. (2022)), the latter would no longer carry the same semantic meaning as *our notion* of a validation set, which is intentionally designed to *not be* from the same distribution as the training set. Instead, our evaluation framework code accesses the *full* dataset via an internal method call to Design Bench, and we construct our own validation set from it. We illustrate these differences in Figure 3.

**Bayesian optimisation** Bayesian optimisation ('BayesOpt') algorithms are typically used in *online* MBO, where the online setting permits access to the ground truth during training, as a way to generate and label new candidates. These algorithms can be seen as sequential decision making processes in which a Bayesian probabilistic model $f_\theta(\boldsymbol{x})$ is used to approximate the ground truth oracle $f$ (e.g. a Gaussian process). Bayesian optimisation alternates between an acquisition function choosing which candidate $\boldsymbol{x}$ to sample next and updating $f_\theta$ to reflect the knowledge gained by having labelled $\boldsymbol{x}$ with ground truth $f$. However, these algorithms require the specification of a prior over the data which can be cumbersome, and therefore recent works have explored the pre-training of such priors on offline data Wang et al. (2021); Hakhamaneshi et al. (2021); Wistuba & Grabocka (2021).

**Learned black box optimisers** Black box optimisation comprises a wide range of algorithms, for instance simulated annealing, genetic algorithms, and Bayesian optimisation (Paragraph 3). Typically these algorithms assume a fixed dimensionality for the input space, making it difficult for them to adapt to different input dimensionalities (for instance among different tasks). Because of this there is interest in learning black box optimisers, with one such instance being 'OptFormer' Chen et al. (2022b) (a transformer-based model). OptFormer is trained on a large corpus of precollected hyperparameter optimisation trajectories for various

---

[6]However, in Trabucco et al. (2022) (their Appendix F) some examples are given as to what validation metrics could be used.

black box algorithms and tasks (i.e. different policies). In principle this can also be utilised for MBO but it would require precollected trajectories from existing policies.

While our focus is on offline MBO, in a real world setting MBO is never *fully* offline; this is because one needs to eventually verify that the candidates generated are actually useful and this can only be done with the ground truth. In other words, offline MBO is a way to 'bootstrap' an online MBO pipeline, by leveraging past oracle evaluations to learn a model which can inform future evaluations. One way we can combine both styles is to consider factorised generative models. For instance, in the case of diffusion models (Section 2.1.1) one can factorise a joint density $p_\theta(\boldsymbol{x}, y)$ into a prior over the data $p_\theta(\boldsymbol{x})$ and a classifier $p_\theta(y|\boldsymbol{x})$. If we assume that classifiers are easier to update in online fashion, then we can train $p_\theta(y|\boldsymbol{x})$ as a Bayesian probabilistic model. At generation time, we can hold the diffusion model fixed and finetune the classifier by performing Bayesian optimisation.

In the more general case however, for generative models that admit latent space encodings of the data, a very straightforward way to leverage knowledge of the former is to train BayesOpt algorithms on those encodings (Maus et al., 2022). This can be especially useful if the latent space is of a much smaller dimension than the data, or if the data space is discrete.

## 3.1 Modelling approaches

**Model inversion networks**  The use of generative models for MBO was proposed by Kumar & Levine (2020), under the name *model inversion networks* (MINs). The name is in reference to the fact that one can learn the *inverse* of the oracle $f_\theta^{-1} : \mathcal{Y} \to \mathcal{X}$, which is a generative model. In their work, GANs are chosen for the generative model, whose model we will denote as $G_\theta(\boldsymbol{z}, y)$ – that is to say: $\boldsymbol{x} \sim p_\theta(\boldsymbol{x}|y)$ implies we sample from the prior $\boldsymbol{z} \sim p(\boldsymbol{z})$, then produce a sample $\boldsymbol{x} = G_\theta(\boldsymbol{z}, y)$. At *generation time* the authors propose the learning of the following prior distribution as a way to extrapolate the generative model[7]:

$$p_\zeta(\boldsymbol{z}, y)^* := \underset{p_\zeta(\boldsymbol{z}, y)}{\arg\max} \ \mathbb{E}_{\boldsymbol{z}, y \sim p_\zeta(\boldsymbol{z}, y)} y + \mathbb{E}_{(\boldsymbol{z}, y) \sim p_\zeta} \Big[ \epsilon_1 \log p_\theta(y|G_\theta(\boldsymbol{z}, y)) + \epsilon_2 \log p(\boldsymbol{z}) \Big], \tag{25}$$

where $\epsilon_1$ and $\epsilon_2$ are hyperparameters that weight the *agreement* and the prior probability of the candidate $\boldsymbol{z}$. The agreement is measuring the log likelihood of $\tilde{\boldsymbol{x}} = G_\theta(\boldsymbol{z}, y)$ being classified as $y$ under the training oracle $f_\theta$ (expressed probabilistically as $p_\theta(y|\boldsymbol{x})$), and can be thought of measuring to what extent the classifier and generative model 'agree' that $\tilde{\boldsymbol{x}}$ has a score of $y$. The log density $p(\boldsymbol{z})$ can be thought of as a regulariser to ensure that the generated candidate $\boldsymbol{z}$ is likely under the latent distribution of the GAN.

We note that agreement can be easily turned into a validation metric by simply substituting $p_\theta(y|\boldsymbol{x})$ for the validation oracle $p_\phi(y|\boldsymbol{x})$. Note that if we assume that $p_\phi(y|\boldsymbol{x})$ is a Gaussian, then the log density of some input $y$ turns into the mean squared error up to some constant terms, so we we can simply write agreement out as $\mathbb{E}_{y \sim p_\gamma(y)} \|f_\phi(G_\theta(\boldsymbol{z}, y)) - y\|^2$. This leads us to our second validation metric, which we formally define as:

$$\boxed{\mathcal{M}_{\text{Agr}}(\mathcal{D}_{\text{valid}}; \theta) = \frac{1}{|\mathcal{D}_{\text{valid}}|} \sum_{(\boldsymbol{x}, y) \sim \mathcal{D}_{\text{valid}}} \|f_\phi(\tilde{\boldsymbol{x}}_i) - y\|^2, \ \ \text{where } \tilde{\boldsymbol{x}}_i \sim p_\theta(\boldsymbol{x}|y)} \tag{26}$$

We remark that Equation 26 has a mathematical connection to the *reverse KL divergence*, one of many divergences used to measure the discrepency between a generative and ground truth distribution. For inclined readers, we provide a derivation expressing this relationship in the appendix, Section A.2.2.

**Discriminative approaches**  In the introduction we noted that MBO methods can be seen as approaching the problem from either a discriminative or a generative point of view (though some overlap can also certainly exist between the two). In the former case, regularising the approximate oracle $f_\theta(\boldsymbol{x})$ is key, and it is also the model that is sampled from (e.g. as per gradient ascent in Equation 3). The key idea is that the approximate

---

[7]In Kumar & Levine (2020) this optimisation is expressed for a single $(\boldsymbol{z}, y)$ pair, but here we formalise it as learning a joint distribution over these two variables. If this optimisation is expressed for a minibatch of $(\boldsymbol{z}, y)$'s, then it can be seen as learning an empirical distribution over those variables.

oracle should act conservatively or pessimistically in out-of-distribution regions. Some examples include mining for and penalising adversarial examples (Trabucco et al., 2021), encouraging model smoothness Yu et al. (2021), conservative statistical estimators such as normalised maximum likelihood (Fu & Levine, 2021), learning bidirectional mappings (Chen et al., 2022a), and mitigating domain shift (Qi et al., 2022).

**Generative approaches**   Brookes et al. (2019) propose the use of variational inference to learn a sampling distribution $p_\theta(\boldsymbol{x}|S)$ given a probabilistic form of the oracle $p_\theta(S|\boldsymbol{x})$ as well as a pre-trained prior distribution over the data $p_\theta(\boldsymbol{x})$. Here, $S$ is some desirable range of $y$'s, and therefore $p_\theta(\boldsymbol{x}|S)$ can be thought of as the 'extrapolated' generative model. In Section A.2.1 we discuss how such a model can be viewed within our evaluation framework. Lastly, Fannjiang & Listgarten (2020) proposes MBO training within the context of a min-max game between the generative model and approximate oracle. Given some target range $y \in S$ an iterative min-max game is performed where the generative model $p_\theta(\boldsymbol{x})$ updates its parameters to maximise the expected conditional probability over samples generated from that range, and the approximate oracle $f_\theta(\boldsymbol{x})$ updates its parameters to minimise the error between the generated predictions and that of the ground truth. Since the latter isn't accessible, an approximation of the error is used instead. In relation to our evaluation framework, the extrapolated model would essentially be the final set of weights $\theta^{(t)}$ for $p_\theta(\boldsymbol{x}|S)|_{\theta=\theta^{(t)}}$ when the min-max game has reached equilibrium.

For both approaches, there is a notion of leveraging an initial generative model $p_\theta(\boldsymbol{x})$ and fine-tuning it with the oracle so that it generates higher-scoring samples in regions that it was not initially trained on. Both the min-max and variational inference techniques can be thought of as creating the 'extrapolated' model within the context of our evaluation framework (Figure 2). Therefore, while our evaluation framework does not preclude these more sophisticated techniques, we have chosen to use the simplest extrapolation technique possible – which is simply switching out the prior distribution – as explained in Section 2.1.

**Diffusion models**   Recently, diffusion models (Ho et al., 2020) have attracted significant interest due to their competitive performance and ease of training. They are also very closely related to score-based generative models (Song & Ermon, 2019; 2020). In diffusion, the task is to learn a neural network that can denoise any $\boldsymbol{x}_t$ to $\boldsymbol{x}_{t-1}$, where $q(\boldsymbol{x}_0, \ldots, \boldsymbol{x}_T)$ defines a joint distribution over increasingly perturbed versions of the real data $q(\boldsymbol{x}_0)$. Assuming that $q(\boldsymbol{x}_T) \approx p(\boldsymbol{x}_T)$ for some prior distribution over $\boldsymbol{x}_T$, to generate a sample Langevin MCMC is used to progressively denoise a prior sample $\boldsymbol{x}_T$ into $\boldsymbol{x}_0$, and the result is a sample from the distribution $p_\theta(\boldsymbol{x}_0)$.[8] To the best of our knowledge, we are not aware of any existing works that evaluate diffusion or score-based generative models on MBO datasets provided by Design Bench, and therefore we consider our exploration into diffusion models here as an orthogonal contribution.

## 4   Experiments and Discussion

**Dataset**   Our codebase is built on top of the Design Bench (Trabucco et al., 2022) framework. We consider all continuous datasets in Design Bench datasets: Ant Morphology, D'Kitty Morphology, Superconductor, and Hopper. Continuous datasets are chosen since we are using Gaussian denoising diffusion models, though discrete variants also exist and we leave this to future work.

Both morphology datasets are ones in which the morphology of a robot must be optimised in order to maximise a reward function. For these datasets, the ground truth oracle is a morphology-conditioned controller. For Superconductor, the ground truth oracle is not accessible and therefore it is approximate. For Hopper, the goal is to sample a large ($\approx 5000$ dimensional) set of weights which are used to parameterise a controller.

**Data splits**   While our framework is built on top of Design Bench, as mentioned in Section 3 the evaluation differs slightly. In Design Bench, the user is only officially prescribed $\mathcal{D}_{\text{train}}$, and any users intending to perform validation or model selection should not use anything external to $\mathcal{D}_{\text{train}}$. However, this is fundamentally incompatible with our evaluation framework since we want our validation set to be out-of-distribution, and subsampling a part of $\mathcal{D}_{\text{train}}$ to create $\mathcal{D}_{\text{valid}}$ does not satisfy this. As illustrated in Figure 3, we break this

---

[8]The Langevin MCMC procedure is theoretically guaranteed to produce a sample from $p_\theta(\boldsymbol{x})$ (Welling & Teh, 2011). As opposed to Equation 3, where no noise is injected into the procedure and therefore samples are mode seeking.

Table 2: Summary of datasets used in this work. †: thresholds shown are all defaults from Design Bench, with the exception of Hopper 50% which is a subset of the original Hopper dataset (see Paragraph 4 for justification); ‡: because no exact oracle exists, the way the dataset is split corresponds to that shown in Figure 3c.

| | # features | $|\mathcal{D}_{\text{train}}|$ / $|\mathcal{D}|$ | $\gamma^\dagger$ | $f$ exists? |
|---|---|---|---|---|
| **Ant Morphology** | 60 | 10004 / 25009 | 165.33 | ✓ |
| **Kitty Morphology** | 56 | 10004 / 25009 | 199.36 | ✓ |
| **Superconductor** | 86 | 17014 / 21263 | 74.0 | ‡✗ |
| **Hopper 50%** | 5126 | 1600 / 3200 | 434.5 | ✓ |

convention and define the validation split to be $\mathcal{D}_{\text{train}} \setminus \mathcal{D}$, i.e. their set difference. Note that *if* a ground truth oracle exists, there is no need to define a $\mathcal{D}_{\text{test}}$, and this is the case for all datasets except Superconductor (Figure 3b). Otherwise, for Superconductor a random 50% subsample of ($\mathcal{D}_{\text{train}} \setminus \mathcal{D}$) is assigned to $\mathcal{D}_{\text{valid}}$ (Figure 3c) and we use the pre-trained test oracle `RandomForest-v0` provided with the framework, which was trained on the full dataset $\mathcal{D}$.

One nuance with the Hopper dataset is that the full dataset $\mathcal{D}$ and the training set $\mathcal{D}_{\text{train}}$ are equivalent, presumably because of the scarcity of examples. This means that a validation set cannot be extracted unless the training set itself is redefined, and this means that the training set in our framework is no longer identical to that originally proposed by Design Bench. To address this, we compute the median $y$ with respect to $\mathcal{D}_{\text{train}}$, and take the lower half as $\mathcal{D}_{\text{train}}$ and the upper half as $\mathcal{D}_{\text{valid}}$. We call the final dataset 'Hopper 50%', to distinguish it from the original dataset.

**Oracle pre-training** The validation oracle $f_\phi$ is an MLP comprising of four hidden layers, trained on $\mathcal{D}_{\text{train}} \cup \mathcal{D}_{\text{valid}}$ with the mean squared error loss function. We do not apply any special regularisation tricks to the model. Note that in the case of the Superconductor – the only dataset that doesn't admit a ground truth oracle $f$ – it is not to be confused with the *approximate test oracle*, which is trained on $\mathcal{D} = \mathcal{D}_{\text{train}} \cup \mathcal{D}_{\text{valid}} \cup \mathcal{D}_{\text{test}}$. The approximate test oracle we use for Superconductor is `RandomForest-v0`, which is provided with the framework.

**Architecture** The architecture that we use is a U-Net derived from HuggingFace's 'annotated diffusion model' [9], whose convolutional operators have been replaced with fully connected layers for all datasets except Hopper. For Hopper, we use 1D convolutions because MLPs performed poorly and significantly blew up the number of learnable parameters. Furthermore, since we know that the Hopper dataset is a feature vector of neural network weights it is useful to exploit locality.

For all experiments we train with the ADAM optimiser (Kingma & Ba, 2014), with a learning rate of $2 \times 10^{-5}$, $\beta = (0.0, 0.9)$, and diffusion timesteps $T = 200$. Experiments are trained for 5000 epochs with single P-100 GPUs. Input data is normalised with the min and max values per feature, with the min and max values computed over the training set $\mathcal{D}_{\text{train}}$. The same is computed for the score variable $y$, i.e. all examples in the training set have their scores normalised to be within $[0, 1]$.

**Experiments** For each validation metric and dataset, we run many experiments where each experiment is a particular instantiation of hyperparameters (see Section A.3.1 for more details), and the experiment is run as per Algorithm 1. By running this algorithm for all experiments we can derive an $m \times N$ matrix $\mathbf{U}$ of validation metric values, where $\mathbf{U}_{ij}$ denotes the best validation metric value found for metric $i$ and experiment $j$. Then, if invoke Algorithm 2 on the same experiments this will give us a matrix $\mathbf{V}$ of $m \times N$ of test rewards. By plotting $\mathbf{U}_i$ against $\mathbf{V}_i$ we obtain the scatterplots shown in Figure 5, and the Pearson correlation can be defined simply as $\text{pearson}(\mathbf{U}_i, \mathbf{V}_i)$ for the $i$'th validation metric.

---

[9] https://huggingface.co/blog/annotated-diffusion

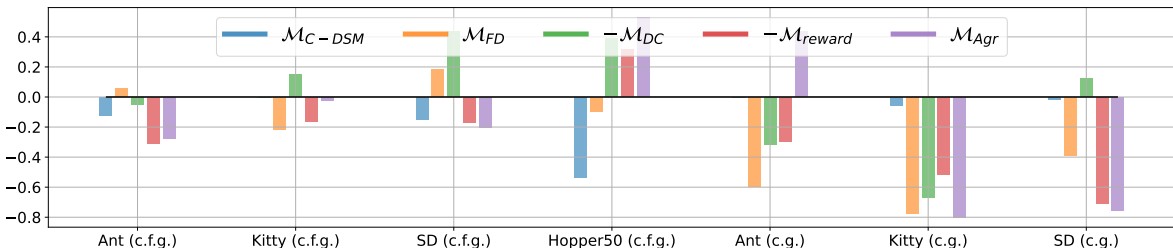

Figure 4: The Pearson correlation computed for each dataset / diffusion variant. sPearson correlations are computed as per the description in Paragraph 4. Since each validation metric is desgned to be minimised, the ideal metric should be highly negatively correlated with the test reward (Equation 24), which is to be maximised. By counting the best metric per experiment, we obtain the following counts (the more ticks the better): $-\mathcal{M}_{reward}$: ✓, $\mathcal{M}_{FD}$: ✓✓, $\mathcal{M}_{Agr}$: ✓✓✓, $\mathcal{M}_{C-DSM}$: ✓

### 4.1 Results

**Classifier-free guidance**   In Figure 4 we plot the Pearson correlations achieved for each dataset as well as each diffusion variant, classifier-free guidance ('cfg') and classifier-based ('cg'). Since all validation metrics are intended to be minimised, we are interested in metrics that correlate the *most negatively* with the final test reward, i.e. the smaller some validation metric is, the larger the average test reward as per Equation 24. We first consider the first four columns of Figure 4, which correspond to just the classifier-free guidance experiments over all the datasets. The two most promising metrics appear to be $\mathcal{M}_{Agr}$ and $\mathcal{M}_{reward}$. Interestingly, $\mathcal{M}_{C\text{-}DSM}$ performs the best for Hopper50, but we also note that this was our worst performing dataset and no we did not obtain favourable results for any validation metric considered. Since we had to modify the dataset to permit a training split, it contains very few examples as well as an unusually large feature to exmaple ratio (i.e. 5126 features for 1600 examples). Since $\mathcal{M}_{C\text{-}DSM}$ is the only validation metric which is *not* a function of the validation oracle, it is possible the validation oracle is too poor for the other metrics to perform well.

Interestingly, $\mathcal{M}_{DC}$ does not perform well for any of the datasets. It is unclear why however, since $\mathcal{M}_{DC}$ is a sum of two terms measuring precision and recall which are both useful things to measure for generative models (see Section A.1). It may require further exploration in the form of weighted sums instead, since it currently is defined to give equal weighting to precision and recall. Since testing out various weighted combinations of the two would have required significantly more compute, we did not explore it.

**Classifier-based guidance**   The last three groups of barplots in Figure 4 constitute the classifier guidance experiments. For this set of experiments the two most promising metrics appear to be $\mathcal{M}_{FD}$ and $\mathcal{M}_{Agr}$. Note that while we have also plotted $\mathcal{M}_{C\text{-}DSM}$ it is not appropriate as a validation metric for classifier-based guidance – this is because during those experiments $\tau$ is deterministically set to 1, which means the score matching loss in Equation 14 is never conditioned on the $y$ variable, and therefore $\epsilon_\theta$ never sees $y$. (The barplots also corroborate this, as the correlation is virtually zero for all experiments.)

**Summarising both guidance variants**   If we count the best validation metric per subplot, then the top three are $\mathcal{M}_{Agr}$ (3 wins), $\mathcal{M}_{FD}$ (2 wins), and $\mathcal{M}_{reward}$ (1 win), respectively. This suggests that if we were to perform a real world MBO then these metrics should be most considered with respect to their rankings. However, since this paper only concerns itself with Gaussian DDPMs, we can not confidently extrapolate this ranking to other classes of generative model or discrete datasets. We leave the testing of discrete diffusion variants to future work.

**Sample quality vs diversity**   We find that the guidance hyperparameter $w$ makes a huge difference to sample quality, and this is shown in Figure 5 for all datasets and across each validation metric. (For brevity's sake, we only show results for classifier-free guidance, and defer the reader to Section A.4 for the equivalent set of plots for classifier guidance.) In each subplot individual points represent experiments and these are also

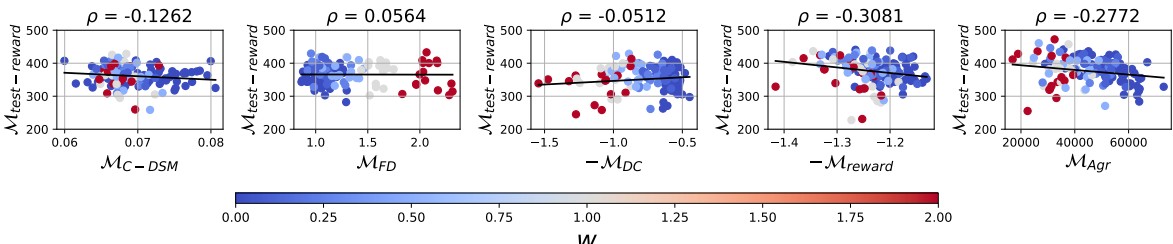

(a) Ant Morphology. $\mathcal{M}_{\mathrm{reward}}$ and $\mathcal{M}_{\mathrm{Agr}}$ are the most negatively correlated with the test reward.

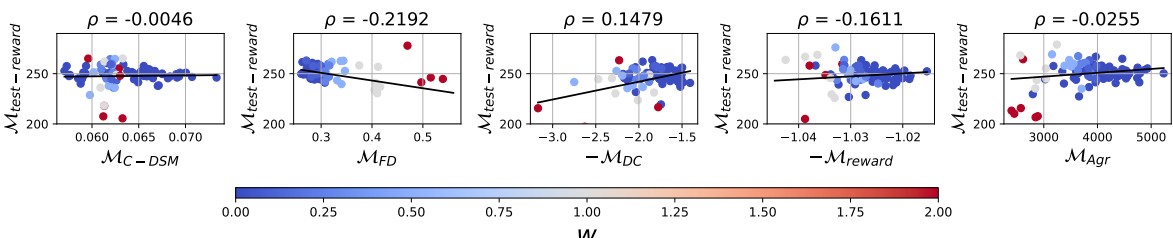

(b) Kitty Morphology (c.f.g.). $\mathcal{M}_{\mathrm{reward}}$ and $\mathcal{M}_{\mathrm{FD}}$ are most negatively correlated with the test reward.

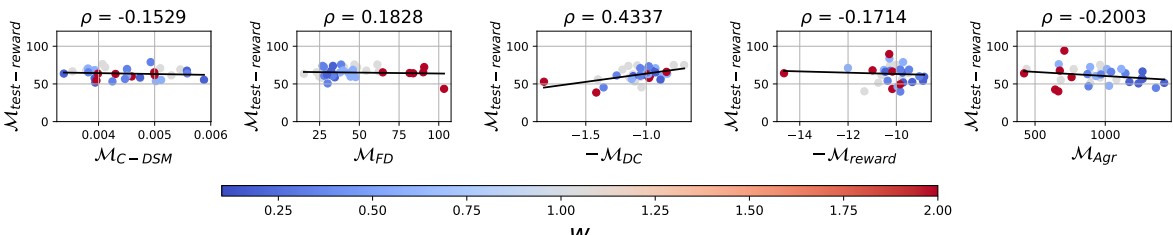

(c) Superconductor (c.f.g.). $\mathcal{M}_{\mathrm{reward}}$ and $\mathcal{M}_{\mathrm{Agr}}$ are most negatively correlated with the test reward.

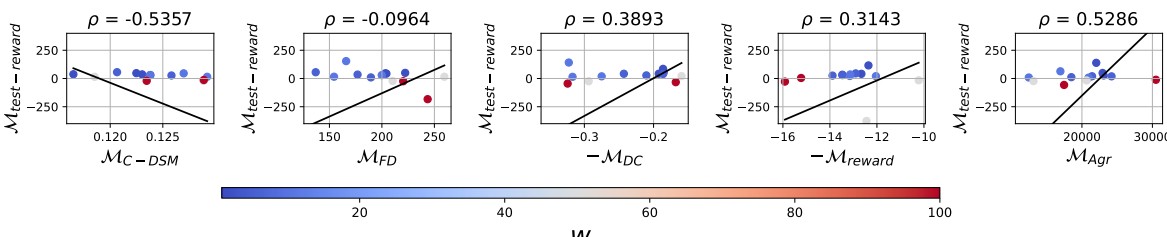

(d) Hopper 50% (c.f.g.). $\mathcal{M}_{\mathrm{FD}}$ and $\mathcal{M}_{\mathrm{C\text{-}DSM}}$ are the most negatively correlated with the test reward.

Figure 5: Correlation plots for each dataset using the classifier-free guidance (c.f.g.) diffusion variant. Each point is colour-coded by $w$, which specifies the strength of the 'implicit' classifier that is derived (Equation 18). We can see that $w$ makes a discernible difference with respect to most of the plots shown. For additional plots for other datasets, please see Section A.4.

colour-coded with guidance hyperparameter $w$. We can see that the choice of $w$ makes a huge difference with respect to the test reward, and appears to highlight a well-established 'dilemma' in generative modelling, which is the trade-off between *sample quality* and *sample diversity* (Ho & Salimans, 2022; Brock et al., 2018; Burgess et al., 2018; Dhariwal & Nichol, 2021). For instance, if sample quality is too heavily weighted, then sample diversity suffers and as a result the candidates we generate – which supposedly comprise high reward – may actually truly be bad candidates when scored by the ground truth. Conversely, if sample diversity

is too heavily weighted then we miss out on modes of the distribution which correspond to high-rewarded candidates.

In Table 3 we present the 100th percentile test rewards for each combination of dataset and conditioning variant. For each of these we simply choose the most promising validation metric as per Figure 4, find the best experiment, and then compute its 100th percentile score as is consistent with evaluation in Design Bench:

$$\mathcal{M}_{100\text{p}}(\mathcal{S}; \theta, \phi) = \max\left[\left\{f\left(\text{sorted}(\mathcal{S}; f_\phi)_i\right)\right\}_{i=1}^K\right], \quad \mathcal{S}_i \sim p_{\theta, \gamma}(\boldsymbol{x}, y) \tag{27}$$

Note that Equation 27 differs from our test reward equation (Equation 24) in that a max is used as the aggregator function instead of the mean. Lastly, we present 50th percentile results in Table S5, which corresponds to a median instead of max.

| | Ant Morphology | D'Kitty Morphology | Superconductor | Hopper50 |
|---|---|---|---|---|
| $\mathcal{D}_{\text{train}}$ | 0.565 | 0.884 | 0.400 | 0.272 |
| **Auto. CbAS** | $0.882 \pm 0.045$ | $0.906 \pm 0.006$ | $0.421 \pm 0.045$ | |
| **CbAS** | $0.876 \pm 0.031$ | $0.892 \pm 0.008$ | $0.503 \pm 0.069$ | |
| **BO-qEI** | $0.819 \pm 0.000$ | $0.896 \pm 0.000$ | $0.402 \pm 0.034$ | |
| **CMA-ES** | $\mathbf{1.214 \pm 0.732}$ | $0.724 \pm 0.001$ | $0.465 \pm 0.024$ | |
| **Grad ascent** | $0.293 \pm 0.023$ | $0.874 \pm 0.022$ | $\mathbf{0.518 \pm 0.024}$ | |
| **Grad ascent (min)** | $0.479 \pm 0.064$ | $0.889 \pm 0.011$ | $0.506 \pm 0.009$ | |
| **Grad ascent (ensemble)** | $0.445 \pm 0.080$ | $0.892 \pm 0.011$ | $0.499 \pm 0.017$ | |
| **REINFORCE** | $0.266 \pm 0.032$ | $0.562 \pm 0.196$ | $0.481 \pm 0.013$ | |
| **MINs** | $0.445 \pm 0.080$ | $0.892 \pm 0.011$ | $0.499 \pm 0.017$ | |
| **COMs** | $0.944 \pm 0.016$ | $\mathbf{0.949 \pm 0.015}$ | $0.439 \pm 0.033$ | |
| **Cond. Diffusion (c.f.g.)** | $\mathbf{0.954 \pm 0.025}$ | $\mathbf{0.972 \pm 0.006}$ | $0.645 \pm 0.115$ | $0.143 \pm 0.037$ |
| **Cond. Diffusion (c.g.)** | $0.929 \pm 0.013$ | $0.952 \pm 0.010$ | $\mathbf{0.664 \pm 0.007}$ | $-$ |

Table 3: 100th percentile test rewards for methods from Design Bench (Trabucco et al., 2022) as well as our diffusion results shown in the last two rows, with c.f.g standing for *classifier-free guidance* (Equation 18) and c.g. standing for *classifier-guidance* (Equation 16). Each result is an average computed over six different runs (seeds). test rewards are min-max normalised with respect to the smallest and largest oracle scores in the *full* dataset, i.e. any scores greater than 1 are *greater* than any score observed in the full dataset. Design Bench results are shown for illustrative purposes only – while our training sets are equivalent to theirs, we use a held-out validation set to guide model selection, which makes a direct comparison to Design Bench difficult.

**Which conditional variant should be used?** From Table 3 both conditioning variants perform roughly on par with each other, though it appears classifier-free guidance performs slightly better. However, they are not necessarily equally convenient to use in a real-world MBO setting. A real-world MBO setting would involve some sort of online learning component since the ground truth oracle needs to be eventually queried (see Section 3, Paragraph 3). Because of this, we recommend the use of classifier-based guidance, where the unconditional generative model $p_\theta(\boldsymbol{x})$ can be independently trained offline while the classifier $p_\phi(y|\boldsymbol{x})$ can be a Bayesian probabilistic model which is able to be updated on per-example basis (i.e. after each query of the ground truth).

## 5  Conclusion

In this work, we asked a fundamental question pertaining to evaluation in offline MBO for diffusion-based generative models: which validation metrics correlate well with the ground truth oracle? The key idea is that if we can run our presented study at scale for a both a difficult and diverse range of datasets for which the ground truth is *known*, insights derived from those findings (such as what are robust validation metrics) can be transferred to more real-world offline MBO tasks where the actual ground truth oracle is expensive to evaluate. To approach this, our evaluation framework is designed to measure how well a generative model extrapolates: the training and validation sets are seen as coming from different $\gamma$-*truncated* distributions, where examples in the validation set comprise a range of $y$'s that are not covered by the training set and are

*larger* than those in the training set. Therefore, from the point of view of the generative model, the validation set is out-of-distribution. Because model selection involves measuring some notion of desirability on the validation set (via a validation metric), we are effectively trying to select for models that can *extrapolate*.

While our proposed evaluation framework is model-agnostic, we presented it in the context of Gaussian DDPMs on four continuous datasets prescribed by Design Bench, as well as across five different validation metrics and two forms of label conditioning for diffusion models: classifier-free and classifier-based guidance. The five validation metrics we chose were inspired by existing MBO works as well as the GAN literature. After ranking the validation metrics based on their correlation with the test reward, we found that the best three performing ones in descending order were agreement, Fréchet Distance, and the validation score, respectively. While we ran a considerable number of experiments in this study, further exploration should be done in testing on discrete datasets (which would require discrete or latent diffusion models) as well as a deeper exploration into how these models perform under different sampling algorithms and design choices (for instance see Karras et al. (2022)).

Lastly, we derived some interesting insights from our work. Regardless of which conditioning variant is used, we found that the most important hyperparameter to tune is the classifier guidance value, which controls the trade-off between sample quality and sample diversity. Furthermore, we posit that the classifier-based variant of diffusion is likely more convenient in practice since it makes it easier to bridge the gap between offline and online MBO.

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

# A Appendix

## A.1 Validation metrics

**Frechet Distance** 'Likelihood-free' metrics are used almost exclusively in the GAN literature because there is no straightforward way to compute likelihoods for this class of models, i.e. $p_\theta(\boldsymbol{x}|y)$ cannot be evaluated. Furthermore, the search for good metrics is still an active topic of research (Borji, 2022). Common likelihood-free metrics involve measuring some distance between distributions in some predefined feature space. For instance, for GANs trained on natural image datasets the *Fréchet Inception Distance* (FID) (Heusel et al., 2017) is used to fit Gaussians to both distributions with respect to the feature space of an InceptionNet classifier trained on ImageNet. Since the acronym 'FID' specifically refers to a very specific InceptionNet-based model, we will simply call it 'FD'. If we assume that FD is computed in some latent space characterised by an arbitrary feature extractor $f_h : \mathcal{X} \to \mathcal{H}$, then FD can be computed in closed form as follows (Dowson & Landau, 1982):

$$\boxed{\mathcal{M}_{\mathrm{FD}}(\boldsymbol{X}, \tilde{\boldsymbol{X}}; f_h) = |\mu(f_h(\boldsymbol{X})) - \mu(f_h(\tilde{\boldsymbol{X}}))| + \mathrm{Tr}(\Sigma(f_h(\boldsymbol{X})) + \Sigma(f_h(\tilde{\boldsymbol{X}})) - 2\Sigma(f_h(\tilde{\boldsymbol{X}}))\Sigma(f_h(\tilde{\boldsymbol{X}}))^{\frac{1}{2}})}$$
(S28a)

where $\mathcal{M}_{\mathrm{FD}} \in \mathbb{R}^+$ and lower FD is better. FD is also known as the 2-Wasserstein distance. Here, $\boldsymbol{X} \in \mathbb{R}^{N \times p}$ denotes $N$ samples coming from a reference distribution (i.e. ground truth) and $\tilde{\boldsymbol{X}}$ are samples coming from the generative model. $\boldsymbol{H} = f_h(\boldsymbol{X})$ denotes these inputs mapped to some feature space. Conveniently, we can simply define the feature space to be with respect to some hidden layer of the *validation oracle*. One caveat of FD is that it may have a stronger bias towards recall (mode coverage) than precision (sample quality) (Kynkäänniemi et al., 2019) and that it reports a single number, which makes it difficult to tease apart how well the model contributes to precision and recall. Furthermore, while there exists a canonical network architecture and set of weights to use for evaluating generative models on natural image datasets (i.e. a particular Inception-V3 network that gives rise to the Frechet *Inception* Distance), this is not the case for other types of datasets. This means that, unless a particular feature extractor is agreed upon, comparing results between papers is non-trivial.

**Density and coverage** We also consider 'density and coverage' (Naeem et al., 2020), which corresponds to an improved version of the 'precision and recall' metric proposed in Kynkäänniemi et al. (2019). In essence, these methods estimate the manifold of both the real and fake data distributions in latent space via the aggregation of hyperspheres centered on each point, and these are used to define precision and recall: precision is the *proportion of fake data that can be explained by real data* (in latent space), and recall is the *proportion of real data that can be explained by fake data* (again, in latent space).

Similar to FD (Paragraph A.1), let us denote $\boldsymbol{H}_i$ as the example $\boldsymbol{X}_i$ embedded in latent space. Let us also define $B(\boldsymbol{H}_i, \mathrm{NND}_K(\boldsymbol{H}_i))$ as the hypersphere centered on $\boldsymbol{H}_i$ whose radius is the $k$-nearest neighbour, and $k$ is a user-specified parameter. 'Density' (the improved precision metric) is defined as:

$$\mathcal{M}_{\mathrm{density}}(\boldsymbol{H}, \tilde{\boldsymbol{H}}; k) = \frac{1}{kN} \sum_{j=1}^{M} \underbrace{\sum_{i=1}^{N} \mathbf{1}\Big\{ \tilde{\boldsymbol{H}}_j \in B(\boldsymbol{H}_i, \mathrm{NND}_k(\boldsymbol{H}_i)) \Big\}}_{\substack{\text{how many real neighbourhoods} \\ \text{does fake sample } \tilde{\boldsymbol{x}}_j \text{ belong to?}}},$$
(S29)

where $\mathbf{1}(\cdot)$ is the indicator function, and large values corresponds to a better density. While coverage (improved 'recall') can be similarly defined by switching around the real and fake terms like so, the authors choose to still leverage a manifold around real samples due to the concern of potentially too many outliers in the

generated distribution $\tilde{\boldsymbol{H}}$. As a result, their coverage is defined as:

$$\mathcal{M}_{\text{coverage}}(\boldsymbol{H}, \tilde{\boldsymbol{H}}; k) = \frac{1}{N} \sum_{i=1}^{N} \bigcup_{j=1}^{M} \mathbf{1}\Big\{ \tilde{\boldsymbol{H}}_j \in B(\boldsymbol{H}_i, \text{NND}_k(\boldsymbol{H}_i)) \Big\} \tag{S30}$$

$$= \frac{1}{N} \sum_{i=1}^{N} \underbrace{\mathbf{1}\Big\{ \exists j \text{ s.t. } \tilde{\boldsymbol{H}}_j \in B(\boldsymbol{H}_i, \text{NND}_k(\boldsymbol{H}_i)) \Big\}}_{\substack{\text{is there } any \text{ fake sample belonging} \\ \text{to } \boldsymbol{x}_i\text{'s neighbourhood?}}}, \tag{S31}$$

where, again, a larger value corresponds to a better coverage. This leads us to the addition of both metrics, $\mathcal{M}_{\text{DC}}$, which is simply:

$$\boxed{\mathcal{M}_{\text{DC}}(\boldsymbol{X}, \tilde{\boldsymbol{X}}; f_h, k) = \mathcal{M}_{\text{density}}(f_h(\boldsymbol{X}), f_h(\tilde{\boldsymbol{X}}); k) + \mathcal{M}_{\text{coverage}}(f_h(\boldsymbol{X}), f_h(\tilde{\boldsymbol{X}}); k)} \tag{S32a}$$

Similar to FD, we use the validation oracle $f_\theta$ to project samples into the latent space. We do not tune $k$ and simply leave it to $k = 3$, which is a recommended default.

## A.2 Related work

### A.2.1 Conditioning by adaptive sampling

CbAS (Brookes et al., 2019), like our proposed method, approaches MBO from a generative modelling perspective. Given some pre-trained 'prior' generative model on the input data $p_\theta(\boldsymbol{x})$, the authors propose the derivation of the conditional generative model $p_\theta(\boldsymbol{x}|y)$ via Bayes' rule:

$$p_\theta(\boldsymbol{x}|y) = \frac{p(y|\boldsymbol{x})p_\theta(\boldsymbol{x})}{p_\theta(y)} = \frac{p(y|\boldsymbol{x})p_\theta(\boldsymbol{x})}{\int_{\boldsymbol{x}} p(y|\boldsymbol{x})p_\theta(\boldsymbol{x})d\boldsymbol{x}}, \tag{S33}$$

where $p(y|\boldsymbol{x})$ denotes the oracle in probabilistic form, and is not required to be differentiable. More generally, the authors use $S$ to denote some target range of $y$'s that would be desirable to condition on, for instance if $p(S|\boldsymbol{x}) = \int_y p(y|\boldsymbol{x})\mathbf{1}_{y \in S}dy$ then:

$$p_\theta(\boldsymbol{x}|S) = \frac{p(S|\boldsymbol{x})p_\theta(\boldsymbol{x})}{p_\theta(S)} = \frac{p(S|\boldsymbol{x})p_\theta(\boldsymbol{x})}{\int_{\boldsymbol{x}} p(S|\boldsymbol{x})p_\theta(\boldsymbol{x})d\boldsymbol{x}}, \tag{S34}$$

Due to the intractability of the denominator term, the authors propose the use of variational inference to learn a sampling network $q_\zeta(\boldsymbol{x})$ that is as close as possible to $p_\theta(\boldsymbol{x}|S)$ as measured by the forward KL divergence. Here, let us use $p_\theta(S|\boldsymbol{x})$ in place of $p(S|\boldsymbol{x})$, and assume the oracle $p_\theta(S|\boldsymbol{x})$ was trained on $\mathcal{D}_{\text{train}}$:[10]

$$\begin{aligned}
\zeta^* &= \arg\min_\zeta \underbrace{\text{KL}\Big[ p_\theta(\boldsymbol{x}|S) \parallel q_\zeta(\boldsymbol{x}) \Big]}_{\text{forward KL}} \\
&= \arg\min_\zeta \sum_{\boldsymbol{x}} p_\theta(\boldsymbol{x}|S) \log\Big( p_\theta(\boldsymbol{x}|S) - q_\zeta(\boldsymbol{x}) \Big) \\
&= \arg\min_\zeta \sum_{\boldsymbol{x}} \Big[ p_\theta(\boldsymbol{x}|S) \log p_\theta(\boldsymbol{x}|S) - p_\theta(\boldsymbol{x}|S) \log q_\zeta(\boldsymbol{x}) \Big] \\
&= \arg\max_\zeta \mathbb{H}[p_\theta(\boldsymbol{x}|S)] + \sum_{\boldsymbol{x}} \Big[ \frac{p_\theta(S|\boldsymbol{x})p_\theta(\boldsymbol{x})}{p_\theta(S)} \log q_\zeta(\boldsymbol{x}) \Big] \\
&= \arg\max_\zeta \underbrace{\mathbb{H}[p_\theta(\boldsymbol{x}|S)]}_{\text{const.}} + \underbrace{\frac{1}{p_\theta(S)}}_{\text{const.}} \sum_{\boldsymbol{x}} \Big[ p_\theta(S|\boldsymbol{x})p_\theta(\boldsymbol{x}) \log q_\zeta(\boldsymbol{x}) \Big] \\
&= \arg\max_\zeta \mathbb{E}_{\boldsymbol{x} \sim p_\theta(\boldsymbol{x})} \Big[ \underbrace{p_\theta(S|\boldsymbol{x})}_{\text{oracle}} \log q_\zeta(\boldsymbol{x}) \Big].
\end{aligned} \tag{S35}$$

---

[10]In their paper the symbol $\phi$ is used, but here we use $\zeta$ since the former is used to denote the validation oracle.

The authors mention that in practice importance sampling must be used for Equation S35. This is because the expectation is over samples in $p_\theta(\boldsymbol{x})$, which in turn was trained on only examples with (relatively) small $y$. i.e. those in $\mathcal{D}_{\text{train}}$. Because of this, $p(S|\boldsymbol{x})$ is likely to be small in magnitude for most samples. For more details, we defer the reader to the original paper (Brookes et al., 2019).

To relate the training of CbAS to our evaluation framework, we can instead consider Equation S35 as part of the *validation* part of our evaluation framework. In other words, if we define $S := [\gamma, \infty]$ and use the validation oracle $p_\phi$ in place of $p_\theta(S|\boldsymbol{x})$, then we can optimise for the *extrapolated model* as the following:

$$\zeta^* = \arg\min_{\zeta} \ \mathbb{E}_{\boldsymbol{x} \sim p_\theta(\boldsymbol{x})}\Big[ \underbrace{p_\phi(S|\boldsymbol{x})}_{\text{oracle}} \log q_\zeta(\boldsymbol{x})\Big] \tag{S36}$$

Generally speaking, validation metrics should not be optimised over directly since they are functions of the validation set, and the purpose of a validation set in turn is to give a less biased measure of generalisation than the same metric computed on the training set. However, this may not be too big of a deal here since we are not taking gradients with respect to the oracle.

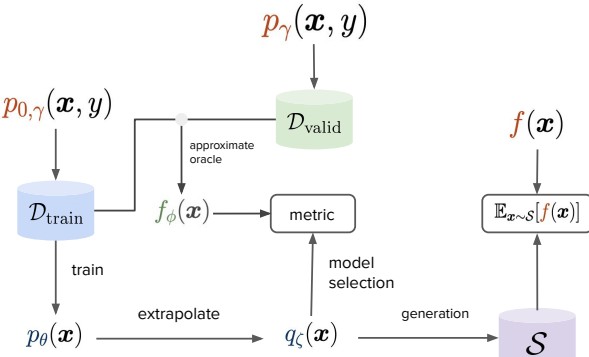

Figure S6: The training and evaluation of CbAS Brookes et al. (2019) in the context of our evaluation framework. The extrapolation equation is described in Equation S35 and involves variational inference to fine-tune $p_\theta(\boldsymbol{x})$ into a search model $q_\zeta(\boldsymbol{x})$.

### A.2.2 Model inversion networks and the reverse KL divergence

It turns out that there is an interesting connection between the agreement and the *reverse KL divergence* between a specific kind of augmented model distribution and the truncated ground truth $p_{0,\gamma}(\boldsymbol{x}, y)$. To see this, let us re-consider the *generation time* optimisation performed in Kumar & Levine (2020) (which we called *MIN-Opt*), which tries to find a good candidate $\boldsymbol{x} = G_\theta(\boldsymbol{z}, y)$ via the following optimisation:

$$y^*, \boldsymbol{z}^* = \arg\max_{y, \boldsymbol{z}} \ y + \epsilon_1 \underbrace{\log p_\theta(y|G_\theta(\boldsymbol{z}, y))}_{\text{agreement}} + \epsilon_2 \underbrace{\log p(\boldsymbol{z})}_{\text{prior over z}} \tag{S37}$$

$\boldsymbol{z}$ and $y$ can be generated by performing gradient ascent with respect to $y$ and $\boldsymbol{z}$. We can also express MIN-Opt with respect to a batch of $(y, \boldsymbol{z})$'s, and this can be elegantly written if we express it as optimising over a *distribution* $p_\zeta(\boldsymbol{z}, y)$. Then we can find such a distribution that maximises the *expected value* of Equation S37 over samples drawn from $p_\zeta(\boldsymbol{z}, y)$:

$$p_\zeta(\boldsymbol{z}, y)^* := \arg\max_{p_\zeta(\boldsymbol{z}, y)} \ \mathbb{E}_{\boldsymbol{z}, y \sim p_\zeta(\boldsymbol{z}, y)}\Big[y + \epsilon_1\Big(\log p_\theta(y|G_\theta(\boldsymbol{z}, y)) + \epsilon_2 \log p(\boldsymbol{z})\Big)\Big], \tag{S38}$$

where e.g. $\zeta$ parameterises the distribution, e.g. a mean and variance if we assume it is Gaussian. Although MIN-Opt was intended to be used at generation time to optimise for good candidates, we can also treat it as a validation metric, especially if we replace the training oracle $p_\theta(y|\boldsymbol{x})$ with the validation oracle $p_\phi(y|\boldsymbol{x})$. For the sake of convenience, let us also replace $\epsilon_1$ and $\epsilon_2$ with one hyperparameter $\eta$. This hyperparameter

can be seen as expressing a trade-off between selecting for large scores $y$ versus ones with large agreement and density under the prior distribution. This gives us the following:

$$p_\zeta(\boldsymbol{z}, y)^* = \underset{p_\zeta(\boldsymbol{z},y)}{\arg\max} \ \mathbb{E}_{y,\boldsymbol{z}\sim p_\zeta(\boldsymbol{z},y)}\Big[y + \eta\Big(\log p_\phi(y|G_\theta(\boldsymbol{z},y)) + \log p(\boldsymbol{z})\Big)\Big]$$

$$= \underset{p_\zeta(\boldsymbol{z},y)}{\arg\max} \ \mathbb{E}_{y\sim p_\zeta(y)}y + \eta\mathbb{E}_{y,\boldsymbol{z}\sim p_\zeta(\boldsymbol{z},y)}\Big[\log p_\phi(y|G_\theta(\boldsymbol{z},y)) + \log p(\boldsymbol{z})\Big] \tag{S39}$$

$$= \underset{p_\zeta(\boldsymbol{z},y)}{\arg\max} \ \mathbb{E}_{y\sim p_\zeta(y)}y + \eta\mathbb{E}_{\boldsymbol{x},y,\boldsymbol{z}\sim p_\theta(\boldsymbol{x}|y,\boldsymbol{z})p_\zeta(\boldsymbol{z},y)}\Big[\log p_\phi(y|\boldsymbol{x}) + \log p(\boldsymbol{z})\Big]. \tag{S40}$$

Note that in the last line we instead use the notation $\boldsymbol{x} \sim p_\theta(\boldsymbol{x}|y,\boldsymbol{z})$ (a delta distribution) in place of $\boldsymbol{x} = G_\theta(\boldsymbol{z},y)$ , which is a deterministic operation. We can show that Equation S39 has a very close resemblence to minimising the reverse KL divergence between a specific kind of augmented model and the $\gamma$-truncated ground truth, with respect to our distribution $p_\zeta(\boldsymbol{z},y)$. Suppose that instead of the typical augmented model $p_{\theta,\gamma}(\boldsymbol{x},y) = p_\theta(\boldsymbol{x}|y)p_\gamma(y)$ we consider one where $\boldsymbol{z}$ and $y$ are drawn from a learnable joint distribution $p_\zeta(\boldsymbol{z},y)$, and we simply denote $p_\theta(\boldsymbol{x}|y,\boldsymbol{z})$ to be a delta distribution (since $\boldsymbol{x} = G_\theta(\boldsymbol{z},y)$ is deterministic). We can write this new augmented model as the following:

$$p_{\theta,\zeta}(\boldsymbol{x},y) = \int_{\boldsymbol{z}} \underbrace{p_\theta(\boldsymbol{x}|y,\boldsymbol{z})}_{\text{GAN}} p_\zeta(\boldsymbol{z},y)d\boldsymbol{z}. \tag{S41}$$

Although this distribution is not tractable, we will only be using it to make the derivations more clear.

Let us work backwards here: if we take Equation S39 but substitute the inner square bracket terms for the *reverse KL* divergence between the augmented model of Equation S41 and the ground truth $p_\gamma(\boldsymbol{x},y)$, we obtain the following:

$$p_\zeta(\boldsymbol{z}, y)^* := \underset{y\sim p_\zeta}{\arg\min} \ -\mathbb{E}_{p_\zeta(y)}y + \eta\underbrace{\text{KL}\Big[p_{\theta,\zeta}(\boldsymbol{x},y) \parallel p_\gamma(\boldsymbol{x},y)\Big]}_{\text{reverse KL}}$$

$$= \underset{p_\zeta}{\arg\min} -\mathbb{E}_{y\sim p_\zeta(y)}y + \eta\Big[\mathbb{E}_{\boldsymbol{x},y\sim p_{\theta,\zeta}(\boldsymbol{x},y)}\log p_{\theta,\zeta}(\boldsymbol{x},y) - \mathbb{E}_{\boldsymbol{x},y\sim p_{\theta,\zeta}(\boldsymbol{x},y)}\log p_\gamma(\boldsymbol{x},y)\Big]$$

$$= \underset{p_\zeta}{\arg\max} \ \mathbb{E}_{y\sim p_\zeta(y)}y - \eta\Big[\mathbb{E}_{\boldsymbol{x},y\sim p_{\theta,\zeta}(\boldsymbol{x},y)}\log p_{\theta,\zeta}(\boldsymbol{x},y) + \mathbb{E}_{\boldsymbol{x},y\sim p_{\theta,\zeta}(\boldsymbol{x},y)}\log p_\gamma(\boldsymbol{x},y)\Big]$$

$$= \underset{p_\zeta}{\arg\max} \ \mathbb{E}_{y\sim p_\zeta(y)}y + \eta\Big[\mathbb{H}[p_{\theta,\zeta}] + \mathbb{E}_{\boldsymbol{x},y\sim p_{\theta,\zeta}(\boldsymbol{x},y)}\log p_\gamma(\boldsymbol{x},y)\Big]$$

$$= \underset{p_\zeta}{\arg\max} \ \mathbb{E}_{y\sim p_\zeta(y)}y + \eta\Big[\underbrace{\mathbb{H}[p_{\theta,\zeta}]}_{\text{entropy}} + \mathbb{E}_{\boldsymbol{x},y,\boldsymbol{z}\sim p_\theta(\boldsymbol{x}|y,\boldsymbol{z})p_\zeta(\boldsymbol{z},y)}\Big[\underbrace{\log p(y|\boldsymbol{x})}_{\text{agreement}} + \log p_\gamma(\boldsymbol{x})\Big]\Big]$$

$$\approx \underset{p_\zeta}{\arg\max} \ \mathbb{E}_{y\sim p_\zeta(y)}y + \eta\Big[\underbrace{\mathbb{H}[p_{\theta,\zeta}]}_{\text{entropy}} + \mathbb{E}_{\boldsymbol{x},y,\boldsymbol{z}\sim p_\theta(\boldsymbol{x}|y,\boldsymbol{z})p_\zeta(\boldsymbol{z},y)}\Big[\underbrace{\log p_\phi(y|\boldsymbol{x})}_{\text{agreement}} + \log p_\gamma(\boldsymbol{x})\Big]\Big]. \tag{S42}$$

The entropy term is not tractable because we cannot evaluate its likelihood. For the remaining two terms inside the expectation, the agreement can be approximated with the validation oracle $p_\phi(y|\boldsymbol{x})$. Howeer, it would not be practical to estimate $\log p_\gamma(x)$ since that would require us to train a separate density $p_\phi(\boldsymbol{x})$ to approximate it.

For clarity, let us repeat Equation S39 here:

$$p_\zeta(\boldsymbol{z}, y)^* = \underset{p_\zeta(\boldsymbol{z},y)}{\arg\max} \ \mathbb{E}_{y\sim p_\zeta(y)}y + \eta\mathbb{E}_{\boldsymbol{x},y,\boldsymbol{z}\sim p_\theta(\boldsymbol{x}|y,\boldsymbol{z})p_\zeta(\boldsymbol{z},y)}\Big[\log p_\phi(y|\boldsymbol{x}) + \log p(\boldsymbol{z})\Big]. \tag{S43}$$

The difference between the two is that (1) there is no entropy term; and (2) $\log p_\gamma(\boldsymbol{x})$ term is replaced with $\log p(\boldsymbol{z})$ for MIN-Opt, which is tractable since the know the prior distribution for the GAN. From these observations, we can conclude that MIN-Opt (S39) comprises an approximation of the reverse KL divergence where the entropy term is omitted and the log density of the *data* is replaced with the log density of the *prior*.

### A.2.3 Exponentially tilted densities

Once the best model has been found via an appropriate validation metric, one can train the same type of model on the full dataset $\mathcal{D}$ using the same hyperparameters as before. Ultimately, we would like to be able to generate candidates whose $y$'s exceed that of the entire dataset, and at the same time at plausible according to our generative model. To control how much we trade-off high likelihood versus high rewarding candidates, we can consider the *exponentially tilted density* (Asmussen & Glynn, 2007; O'Donoghue et al., 2020; Piche et al., 2022):

$$p_{\theta,\gamma}(\boldsymbol{x}, y) \exp(\eta^{-1} y - \kappa(\eta)), \tag{S44}$$

where $\kappa(\eta)$ is a normalisation constant, and smaller $\eta$ puts larger emphasis on sampling from regions where $y$ is large. Taking the log of Equation S44, we arrive at:

$$\boldsymbol{x}^*, y^* = \underset{\boldsymbol{x}, y}{\arg\max} \ \log p_{\theta,\gamma}(\boldsymbol{x}, y) + \frac{1}{\eta} y, \tag{S45}$$

In practice, it would not be clear what the best $\eta$ should be, but a reasonable strategy is to consider a range of $\eta$'s, where larger values encode a higher tolerance for 'risk' since these values favour higher rewarding candidates at the cost of likelihood. Note that for VAEs and diffusion models, $\log p_\theta(\boldsymbol{x}, y)$ will need to be approximated with the ELBO. Interestingly, since diffusion models have an extremely close connection to score-based models, one could 'convert' a diffusion model to a score-based model (Weng, 2021) and derive $\nabla_{\boldsymbol{x}, y} \log p_\theta(\boldsymbol{x}, y)$, and this would make sampling trivial.

One potential issue however relates to our empirical observation that predicted scores for generated candidates exhibit very high variance, i.e. the agreement scores are very high (see Figures S7b and S9b). In other words, when we sample some $\boldsymbol{x}, y \sim p_{\theta,\gamma}(\boldsymbol{x}, y)$ (i.e. from the *augmented model*) there is significant uncertainty as to whether $\boldsymbol{x}$ really does have a score of $y$. One potential remedy is to take inspiration from the MIN-Opt generation procedure (Section A.2.2) and add the agreement term to Equation S45:

$$\boldsymbol{x}^*, y^* = \underset{\boldsymbol{x}, y}{\arg\max} \ \log p_{\theta,\gamma}(\boldsymbol{x}, y) + \frac{1}{\eta} y + \alpha \log p_\phi(y|\boldsymbol{x}). \tag{S46}$$

Due to time constraints, we leave additional experimentation here to future work.

### A.3 Additional training details

### A.3.1 Hyperparameters

The architecture that we use is a convolutional U-Net from HuggingFace's 'annotated diffusion model' [11], whose convolutional operators have been replaced with fully connected layers (since Ant and Kitty morphology inputs are flat vectors).

For all experiments we train with the ADAM optimiser (Kingma & Ba, 2014), with a learning rate of $2 \times 10^{-5}$, $\beta = (0.0, 0.9)$, and diffusion timesteps $T = 200$. Experiments are trained for 5000 epochs with single P-100 GPUs. Input data is normalised with the min and max values per feature, with the min and max values computed over the training set $\mathcal{D}_{\text{train}}$. The same is computed for the score variable $y$, i.e. all examples in the training set have their scores normalised to be within $[0, 1]$.

Here we list hyperparameters that differ between experiments:

- `diffusion_kwargs.tau`: for classifier-free diffusion models, this is the probability of dropping the label (score) $y$ and replacing it with a null token. For classifier guidance models, this is fixed to $\tau = 1$ since this would correspond to training a completely unconditional model.

- `gen_kwargs.dim`: channel multiplier for U-Net architecture

- `diffusion_kwargs.w_cg`: for classifier-based guidance, this is the $w$ that corresponds to the $w$ in Equation 16.

---

[11] https://huggingface.co/blog/annotated-diffusion

### A.3.2 Hyperparameters explored for classifier-free guidance

```
{
    'diffusion_kwargs.tau': {0.05, 0.1, 0.2, 0.4, 0.5},
    'gen_kwargs.dim': {128, 256}
}
```

### A.3.3 Hyperparameters explored for classifier guidance

```
{
    'diffusion_kwargs.w_cg': {1.0, 10.0, 100.0},
    'epochs': {5000, 10000},
    'gen_kwargs.dim': {128, 256}
}
```

### A.3.4 Classifier guidance derivation

Let us denote $\boldsymbol{x}_t$ as the random variable from the distribution $q(\boldsymbol{x}_t)$, denoting noisy input $\boldsymbol{x}$ at timestep $t$. Through Bayes' rule we know that $q(\boldsymbol{x}_t|y) = \frac{q(\boldsymbol{x}_t,y)}{q(y)} = \frac{q(y|\boldsymbol{x}_t)q(\boldsymbol{x}_t)}{q(y)}$. Taking the score $\nabla_{\boldsymbol{x}_t} \log q(\boldsymbol{x}_t|y)$ (which does not depend on $q(y)$), we get:

$$\nabla_{\boldsymbol{x}_t} \log q(\boldsymbol{x}_t|y) = \nabla_{\boldsymbol{x}_t} \log q(y|\boldsymbol{x}_t) + \nabla_{\boldsymbol{x}_t} \log q(\boldsymbol{x}_t) \tag{S47}$$

$$\approx \frac{-1}{\sqrt{1-\bar{\alpha}_t}} \Big( \epsilon_\theta(\boldsymbol{x}_t, t, y) - \epsilon_\theta(\boldsymbol{x}_t, t) \Big), \tag{S48}$$

where in the last line we make clear the connection between the score function and the noise predictor $\epsilon_\theta$ Weng (2021). Since we would like to derive the conditional score, we can simply re-arrange the equation to obtain it:

$$\epsilon_\theta(\boldsymbol{x}_t, t, y) = \epsilon_\theta(\boldsymbol{x}_t, t) - \sqrt{1-\bar{\alpha}_t} \nabla_{\boldsymbol{x}_t} \log q(y|\boldsymbol{x}_t) \tag{S49}$$

$$\approx \epsilon_\theta(\boldsymbol{x}_t, t) - \sqrt{1-\bar{\alpha}_t} \nabla_{\boldsymbol{x}_t} \log p_\theta(y|\boldsymbol{x}_t), \tag{S50}$$

where we approximate the classifier $q(y|\boldsymbol{x}_t)$ with our (approximate) training oracle $p_\theta(y|\boldsymbol{x}_t)$. In practice, we can also define the weighted version as follows, which allows us to balance between conditional sample quality and sample diversity:

$$\epsilon_\theta(\boldsymbol{x}_t, t, y; w) = \epsilon_\theta(\boldsymbol{x}_t, t) - \sqrt{1-\bar{\alpha}_t} w \nabla_{\boldsymbol{x}_t} \log p_\theta(y|\boldsymbol{x}_t), \tag{S51}$$

Therefore, in order to perform classifier-guided generation, we replace $\epsilon_\theta(\boldsymbol{x}_t, t)$ in whatever generation algorithm we use with $\epsilon_\theta(\boldsymbol{x}_t, t, y; w)$ instead.

### A.3.5 Classifier-free guidance

In classifier-free guidance a conditional score estimator $\epsilon_\theta(\boldsymbol{x}_t, y, t)$ is estimated via the algorithm described in Ho & Salimans (2022), where the $y$ token is dropped during training according to some probability $\tau$. If $y$ is dropped it is replaced with some unconditional token. In other words, the noise predictor (score estimator) is trained both conditionally and unconditionally, which means we have both $\epsilon_\theta(\boldsymbol{x}_t, t)$ as well as $\epsilon_\theta(\boldsymbol{x}_t, y, t)$.

From Bayes' rule, we know that: $p(y|\boldsymbol{x}_t) = \frac{p(y, \boldsymbol{x}_t)}{p(\boldsymbol{x}_t)} = \frac{p(\boldsymbol{x}_t|y)p(y)}{p(\boldsymbol{x}_t)}$, and that therefore the score $\nabla_{\boldsymbol{x}_t} \log p(y|\boldsymbol{x}_t)$ is:

$$\nabla_{\boldsymbol{x}_t} \log p(y|\boldsymbol{x}_t) = \nabla_{\boldsymbol{x}_t} \log p(\boldsymbol{x}_t|y) - \nabla_{\boldsymbol{x}_t} \log p(\boldsymbol{x}_t) \tag{S52}$$

We simply plug this into Equation 16 to remove the dependence on $p_\theta(y|\boldsymbol{x}_t)$:

$$\epsilon_\theta(\boldsymbol{x}_t, t, y; w) = \epsilon_\theta(\boldsymbol{x}_t, t) - \sqrt{1 - \bar{\alpha}_t} w \nabla_{\boldsymbol{x}_t} \log p_\theta(y|\boldsymbol{x}_t) \tag{S53}$$

$$= \epsilon_\theta(\boldsymbol{x}_t, t) - \sqrt{1 - \bar{\alpha}_t} w \left[ \nabla_{\boldsymbol{x}_t} \log p_\theta(\boldsymbol{x}_t|y) - \nabla_{\boldsymbol{x}_t} \log p_\theta(\boldsymbol{x}_t) \right] \tag{S54}$$

$$= \epsilon_\theta(\boldsymbol{x}_t, t) - \sqrt{1 - \bar{\alpha}_t} w \left[ \frac{-1}{\sqrt{1 - \bar{\alpha}_t}} \epsilon_\theta(\boldsymbol{x}_t, y, t) - \frac{-1}{\sqrt{1 - \bar{\alpha}_t}} \epsilon_\theta(\boldsymbol{x}_t, t) \right] \tag{S55}$$

$$= \epsilon_\theta(\boldsymbol{x}_t, t) + w \epsilon_\theta(\boldsymbol{x}_t, y, t) - w \epsilon_\theta(\boldsymbol{x}_t, t) \tag{S56}$$

$$= \epsilon_\theta(\boldsymbol{x}_t, t) + w \Big( \epsilon_\theta(\boldsymbol{x}_t, y, t) - \epsilon_\theta(\boldsymbol{x}_t, t) \Big) \tag{S57}$$

### A.4 Correlation and agreement plots

**Agreement plots** We plot the conditioning $y \in \text{linspace}(y_{\min}, y_{\max})$ against the validation/test oracle predictions for candidates conditionally generated with that $y$. We call these 'agreement plots' since the sum of squared residuals for each point would constitute the agreement (with a perfect agreement of zero corresponding to a diagonal dotted line on each graph). Here we demonstrate this amongst the best three models *with respect to* $\mathcal{M}_{\text{Agr}}$, since this is most correlated with the test oracle (see Figure S7a). The shaded regions denote $\pm 1$ standard deviation from the mean, and the marker symbols denote the max/min score for each $y$ with respect to either oracle.

### A.4.1 Ant Morphology

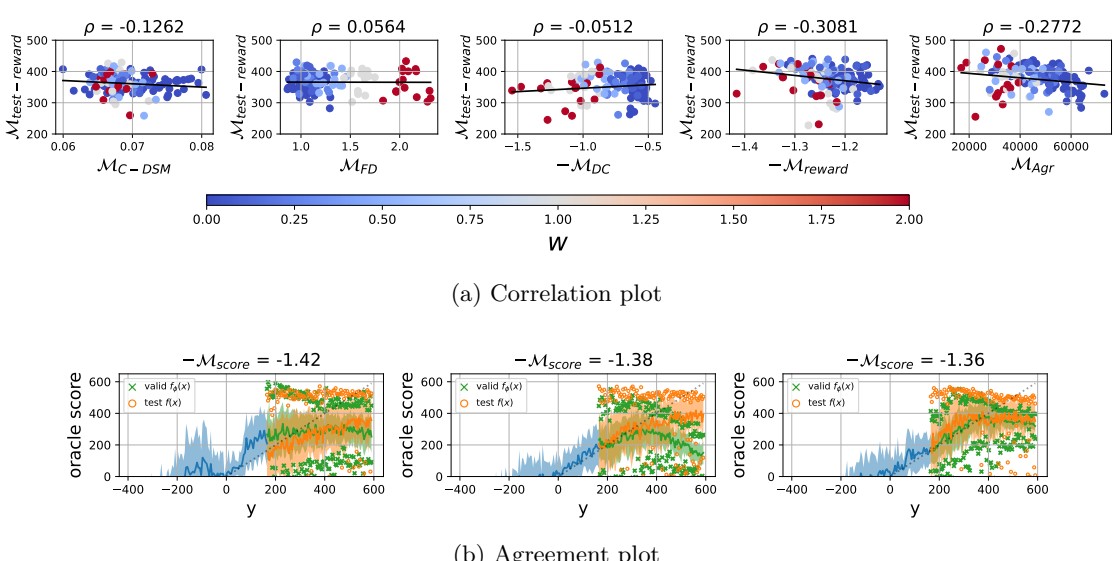

(a) Correlation plot

(b) Agreement plot

Figure S7: Results on Ant Morphology dataset, using the classifier-free guidance variant (Equation 18).

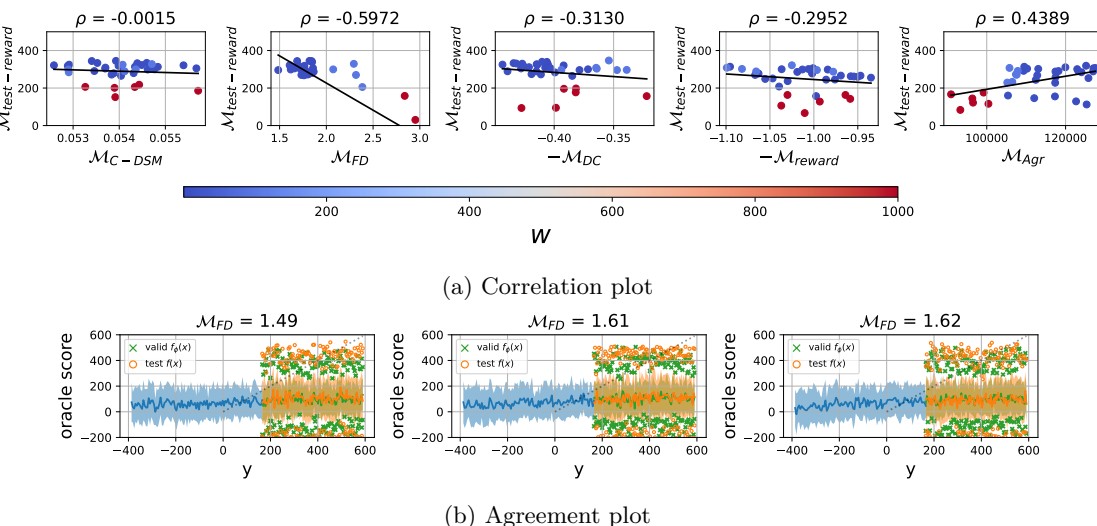

(a) Correlation plot

(b) Agreement plot

Figure S8: Results on Ant Morphology dataset, using the classifier guidance variant (Equation 16).

### A.4.2 D'Kitty Morphology

See Figures S9 and S10.

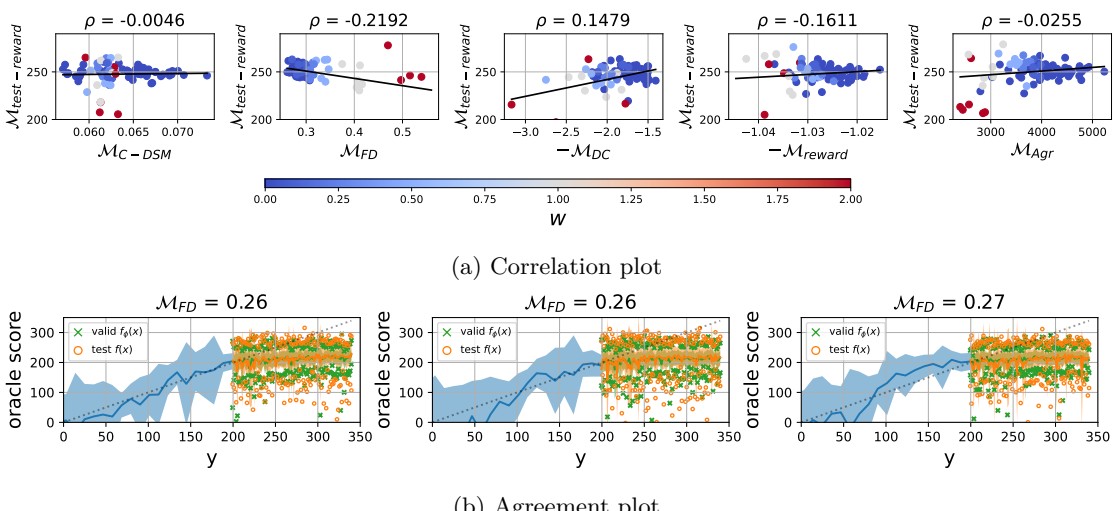

(a) Correlation plot

(b) Agreement plot

Figure S9: Results on D'Kitty Morphology dataset, using the classifier-free guidance variant (Equation 18).

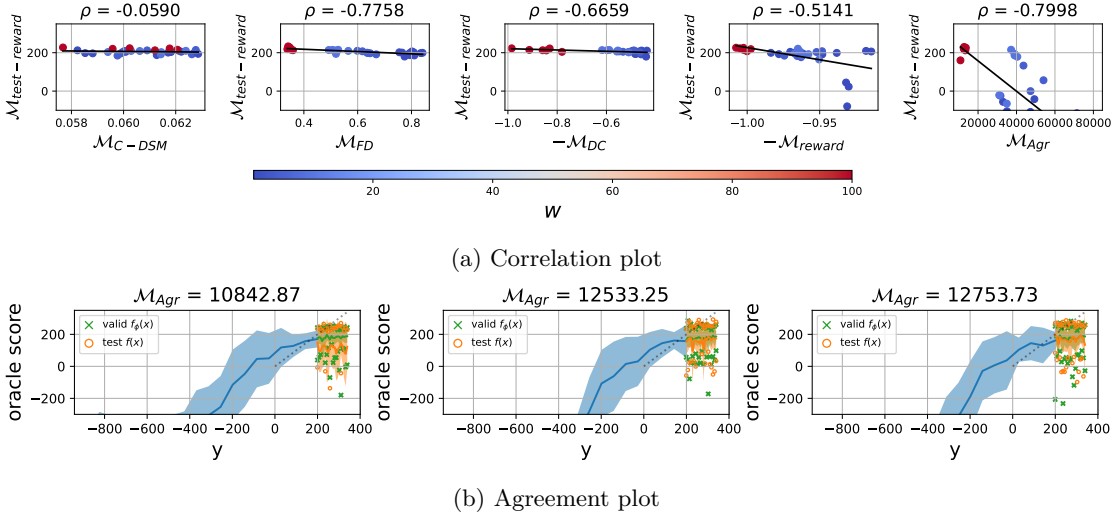

(a) Correlation plot

(b) Agreement plot

Figure S10: Results on D'Kitty Morphology dataset, using the classifier guidance variant (Equation 16).

### A.4.3 Superconductor

See Figure S11.

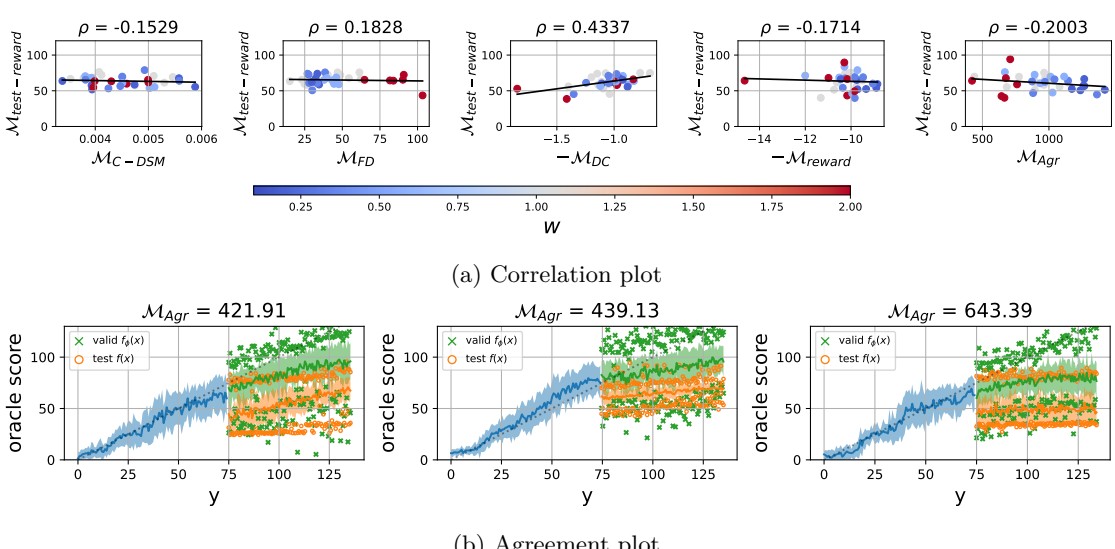

(a) Correlation plot

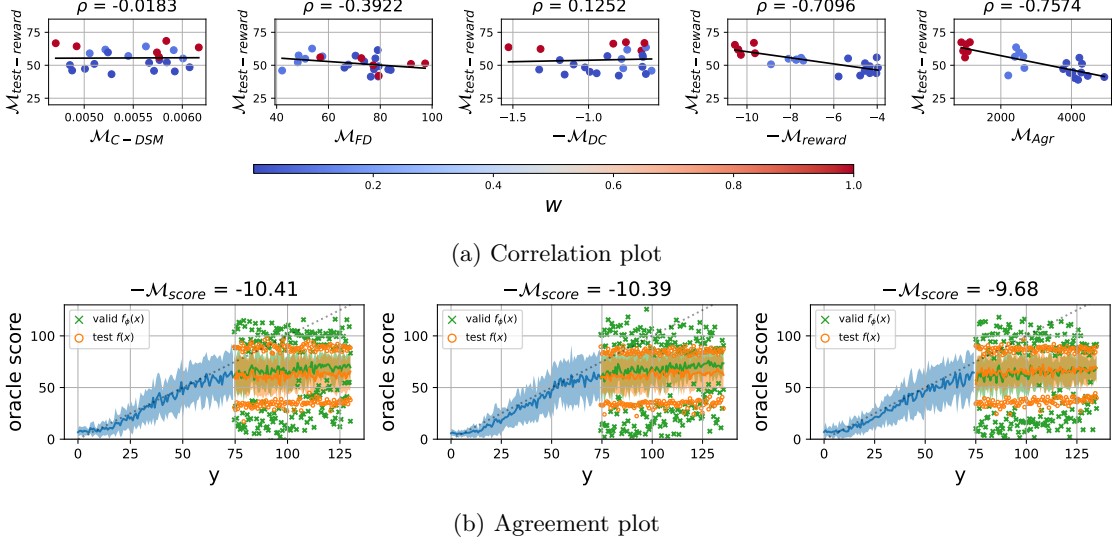

(b) Agreement plot

Figure S11: Results on Superconductor dataset (Hamidieh, 2018), using the classifier-free guidance variant (Equation 18).

(a) Correlation plot

(b) Agreement plot

Figure S12: Results on Superconductor dataset (Hamidieh, 2018), using the classifier-free guidance variant (Equation 18).

|  | 50th pt. | 100th pt. |
|---|---|---|
| $\mathcal{D}_{\text{train}}$ | − | 0.400 |
| **Auto. CbAS** | $0.131 \pm 0.010$ | $0.421 \pm 0.045$ |
| **CbAS** | $0.111 \pm 0.017$ | $0.503 \pm 0.069$ |
| **BO-qEI** | $0.300 \pm 0.015$ | $0.402 \pm 0.034$ |
| **CMA-ES** | $0.379 \pm 0.003$ | $0.465 \pm 0.024$ |
| **Grad.** | $0.476 \pm 0.022$ | $\mathbf{0.518 \pm 0.024}$ |
| **Grad. Min** | $\mathbf{0.471 \pm 0.016}$ | $0.506 \pm 0.009$ |
| **Grad. Mean** | $0.469 \pm 0.022$ | $0.499 \pm 0.017$ |
| **REINFORCE** | $0.463 \pm 0.016$ | $0.481 \pm 0.013$ |
| **MINs** | $0.336 \pm 0.016$ | $0.499 \pm 0.017$ |
| **COMs** | $0.386 \pm 0.018$ | $0.439 \pm 0.033$ |
| **Cond. Diffusion (c.f.g.)** | $\mathbf{0.518 \pm 0.045}$ | $\mathbf{0.636 \pm 0.034}$ |

Table S4: 100th and 50th percentile test rewards for the Superconductor dataset. Results above our conditional diffusion results (bottom-most row) were extracted from Design Bench Trabucco et al. (2022). For our experiments, each result is an average computed over three different runs (random seeds). test rewards are min-max normalised with respect to the smallest and largest oracle scores in the *full* dataset, i.e. any scores greater than 1 are *greater* than any score observed in the full dataset. Design Bench results are shown for illustrative purposes only, and are not directly comparable to our results due to differences in evaluation setup.

### A.4.4 Hopper (50%)

See Figure S13.

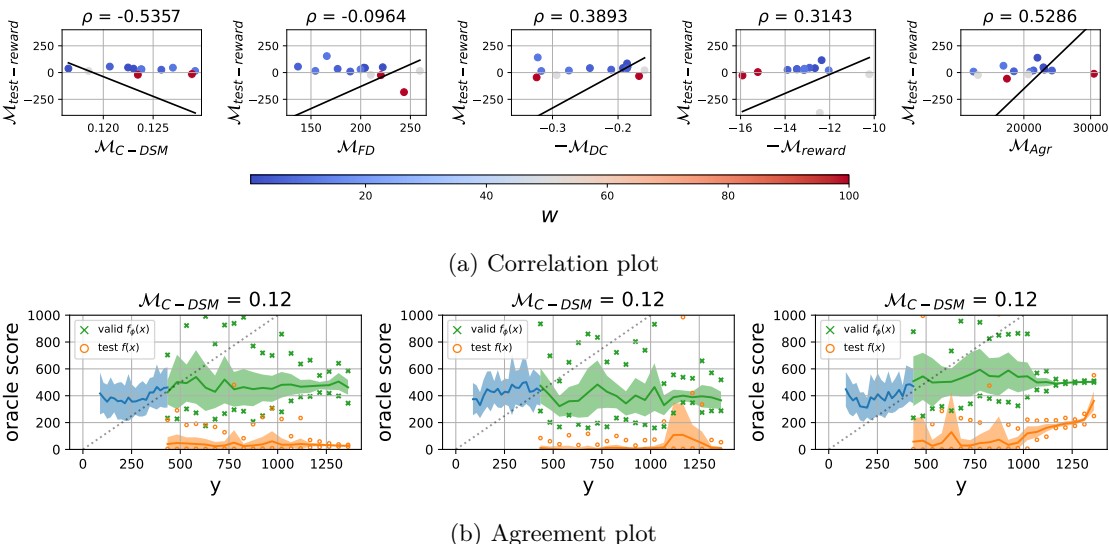

(a) Correlation plot

(b) Agreement plot

Figure S13: Results on Hopper 50%, using the classifier-free guidance variant (Equation 18).

## A.5 Sensitivity to $\tau$ hyperparameter

In Figure S14 we present the same correlation plots as shown in Section A.4 but colour-coded with respect to $\tau$, which is the classifier-free guidance hyperparameter that controls the dropout probability for the label $y$. For each dataset, for the validation metrics that perform the best (i.e. correlate most negatively with the test reward), smaller values of $\tau$ result in better test rewards. Overall, the results indicate that $\tau$ is a sensitive hyperparameter and should be carefully tuned.

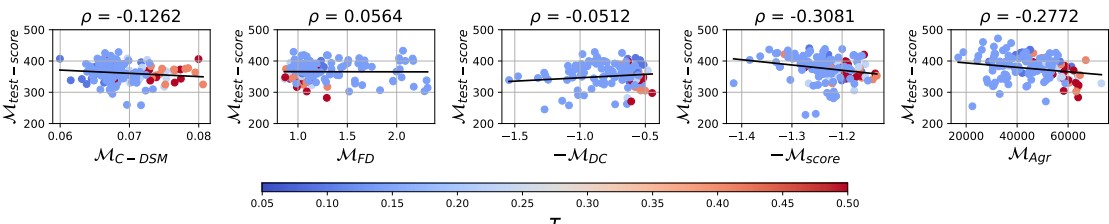

(a) Ant Morphology dataset. For each validation metric, we plot each experiment's smallest-achieved metric versus the test reward (Equation 24). The colourbar represents values of $\tau$.

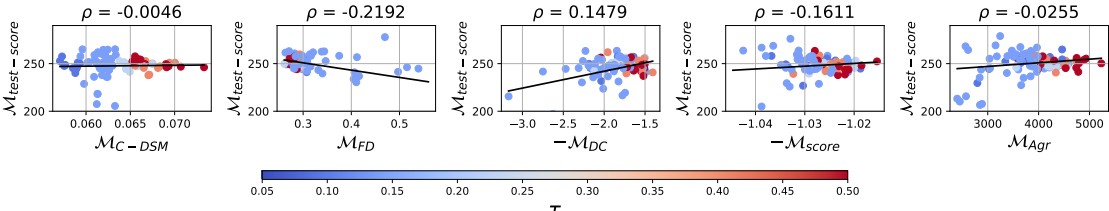

(b) Kitty dataset. For each validation metric, we plot each experiment's smallest-achieved metric versus the test reward (Equation 24). The colourbar represents values of $\tau$.

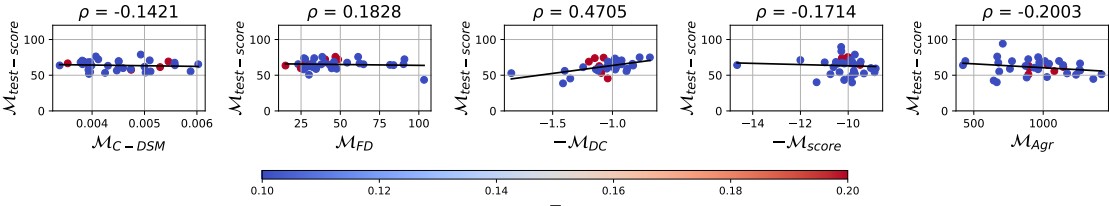

(c) Superconductor dataset. For each validation metric, we plot each experiment's smallest-achieved metric versus the test reward (Equation 24). The colourbar represents values of $\tau$.

Figure S14: Results on each dataset, using the classifier-free guidance variant (Equation 18). Coloured points represent the hyperparameter $\tau$, which represents the dropout probability for classifier-free guidance (cfg). Across all datasets, smaller values of $\tau$ correspond to better scores. This suggests that experiments are quite sensitive to the value of this hyperparameter.

## A.6 Additional results

| | Ant Morphology | D'Kitty Morphology | Superconductor |
|---|---|---|---|
| **Auto. CbAS** | $0.364 \pm 0.014$ | $0.736 \pm 0.025$ | $0.131 \pm 0.010$ |
| **CbAS** | $0.384 \pm 0.016$ | $0.753 \pm 0.008$ | $0.017 \pm 0.503$ |
| **BO-qEI** | $0.567 \pm 0.000$ | $0.883 \pm 0.000$ | $0.300 \pm 0.015$ |
| **CMA-ES** | $-0.045 \pm 0.004$ | $0.684 \pm 0.016$ | $0.379 \pm 0.003$ |
| **Gradient Ascent** | $0.134 \pm 0.018$ | $0.509 \pm 0.200$ | $\mathbf{0.476 \pm 0.022}$ |
| **Grad. Min** | $0.185 \pm 0.008$ | $0.746 \pm 0.034$ | $0.471 \pm 0.016$ |
| **Grad. Mean** | $0.187 \pm 0.009$ | $0.748 \pm 0.024$ | $0.469 \pm 0.022$ |
| **MINs** | $\mathbf{0.618 \pm 0.040}$ | $\mathbf{0.887 \pm 0.004}$ | $0.336 \pm 0.016$ |
| **REINFORCE** | $0.138 \pm 0.032$ | $0.356 \pm 0.131$ | $0.463 \pm 0.016$ |
| **COMs** | $0.519 \pm 0.026$ | $0.885 \pm 0.003$ | $0.386 \pm 0.018$ |
| **Cond. Diffusion (c.f.g.)** | $0.831 \pm 0.052$ | $0.930 \pm 0.004$ | $0.492 \pm 0.112$ |
| **Cond. Diffusion (c.g.)** | $0.880 \pm 0.012$ | $0.935 \pm 0.006$ | – |

Table S5: 50th percentile test rewards for methods from Design Bench (Trabucco et al., 2022) as well as our diffusion results shown in the last two rows, with c.f.g standing for *classifier-free guidance* (Equation 18) and c.g. standing for *classifier-guidance* (Equation 16). Each result is an average computed over six different runs (seeds). test rewards are min-max normalised with respect to the smallest and largest oracle scores in the *full* dataset, i.e. any scores greater than 1 are *greater* than any score observed in the full dataset. Design Bench results are shown for illustrative purposes only, and are not directly comparable to our results due to differences in evaluation setup.

