# OpenReview forum: "Exploring validation metrics for offline model-based optimisation with diffusion models"
_TMLR — Accepted by TMLR_

### Review · Reviewer_1kLh · 2023-07-07

**Summary Of Contributions:**

This paper explores various types of validation metrics for model based optimization. Model based optimization involves generating high scoring designs by learning a model on the existing data. This paper focusses on the offline setting operating on a fixed dataset, and generative models are used to train on the data.

For MBO to be able to successfully generate high scoring examples, the trained model must be able to extrapolate to high score designs while training on a dataset of low score designs. This is achieved by splitting the data set into train, validation and test sets based on the scores. The model extrapolates by replacing the prior of the generative model.

The validation oracle is trained on the train and test sets, and is used in the validation metrics proposed in the paper. The purpose of validation is to evaluate the extrapolation performance of the model. The model hyper-parameters are selected using the validation metric. The final generated designs are evaluated on the test set for the final score.

The methods proposed in this paper are tested on datasets from design bench, and the models used are denoising diffusion probabilistic models.

**Audience:**

Yes

**Broader Impact Concerns:**

No concerns.

**Claims And Evidence:**

Yes

**Requested Changes:**

- A complete algorithm block showing all the steps of the proposed approach is missing. Is the final algorithm as simple as this?
  1. Train a validation oracle on D_train + D_val
  2. Train a DDPM model on D_train and use the validation oracle to select the best hyper-parameters.
  3. Return highest scoring designs using the trained model.

### Suggestions:
- The organization and flow of the paper needs significant improvement.
- Related ideas can be clustered together so that it is easier for the reader to compare them and also find them when referring back.
- Section 1.1 needs significant improvement to provide a strong first impression of the main idea of the paper.
- Figure 1 could be moved to Section 1.1 for easier understanding.

**Strengths And Weaknesses:**

### Strengths
- The proposed approach seems to be practical and well motivated.
- A reliable validation function can significantly speed up modeling iterations and exploration, as such the problem is worth studying.
- Results on benchmark experiments show improved final test scores on the considered datasets.

### Weaknesses
- The paper is not well organized nor well written. It was hard to understand the setup of the paper until I read the whole paper once and re-read again. Figure 1 on the first page for example doesn't add much to the understanding of the reader. In fact, using the notation before introducing it leads to confusion.
- Validation metrics are scattered across the paper, making the paper hard to follow.
- The 'transfer' approach while sounding reasonable, is left to future work - even though that is the main premise of this paper. Furthermore, the benefits of the 'transfer' approach in Section 1.1 are very unclear until one has read the whole paper.
- The novelty in the exploration approach is not clear. It is unclear if the notation $p_\gamma$ and $p_{0, \gamma}$ play any role at all in the proposed method. Is this notation introduced for describing the method, or are they actually modeled in the approach?

---

### Review · Reviewer_qSSg · 2023-07-12

**Summary Of Contributions:**

The paper explores offline model-based optimization. The objective of this research is to initially train a model using a collection of observations from the objective function of interest. This objective is assumed to be too costly to optimize directly. Instead, the final solution is obtained by optimizing directly model with, for example, gradient descent.

The main contribution of the paper is to compare different evaluation strategies for model-based optimization approaches when the real objective is inaccessible, and we can only divide the available datasets. Rather than solely validating the model using hold-out validation data, as is typically done in standard supervised learning, the proposed evaluation process emphasizes the accuracy of the model in regions of the input space that are likely to contain the optimum.

**Audience:**

Yes

**Broader Impact Concerns:**

no ethical concerns

**Claims And Evidence:**

Yes

**Requested Changes:**

I believe the paper would be strengthened if the authors address the inconclusive results of the experiments. Additionally, I think the paper would benefit if the authors could expand the related work section, as mentioned above.


**Strengths And Weaknesses:**


## Strengths

- Overall, the paper is very well written. I particularly appreciate the color coding, as it helps to disambiguate different terms and to better follow the paper.

- The paper nicely motivates the proposed validation metrics.

- The empirical evaluation overall seems sound.


## Weaknesses


**Related work**

I would appreciate a more comprehensive discussion on the relationship between offline model-based optimization and the literature on experimental design or Bayesian optimization. Although I acknowledge that the setting differs because we assume no access to the true objective, recent research on transfer Bayesian optimization or zero-shot optimization also explores scenarios where models are initially trained on offline data. I believe that both fields can mutually benefit from each other, and the paper would become stroner by incorporating a discussion on these related areas.

*Pre-trained Gaussian processes for Bayesian optimization*
Zi Wang, George E. Dahl, Kevin Swersky, Chansoo Lee, Zachary Nado, Justin Gilmer, Jasper Snoek, Zoubin Ghahramani

*Towards Learning Universal Hyperparameter Optimizers with Transformers*
Yutian Chen, Xingyou Song, Chansoo Lee, Zi Wang, Qiuyi Zhang, David Dohan, Kazuya Kawakami, Greg Kochanski, Arnaud Doucet, Marc'aurelio Ranzato, Sagi Perel, Nando de Freitas

*Few-Shot Bayesian Optimization with Deep Kernel Surrogates*
Martin Wistuba, Josif Grabocka




**Inconclusive results**

Looking at Figure 3, the results seem somewhat inconclusive. In certain cases, the validation scores do not correlate significantly with the underlying ground truth. For instance, the MDC exhibits close to zero correlation on the ANT benchmark when used with a cfg model, while displaying a high correlation with a cg model. This inconsistency makes it challenging to rely on these scores for model selection.

**Limited Experiments**

So far the paper considers only a single model type. I understand that experiments can always be extended, but the computation budget might not allow that. However, at this point it unclear to me how much these validation metric rely under the underlying model approach.


**Nitpicks:**

- Equation 5,6 should be max instead of min?


**Questions:**

- How sensitivity are the presented results to hyperparameter  \tau?

- More high level question: I am wondering if also the uncertainty of the model should be taken into account for the validation metric. In other words how certain is the model in it's prediction of the true objective function.

---

### Review · Reviewer_5yiS · 2023-10-16

**Summary Of Contributions:**

The paper claims to
*) propose a conceptual evaluation framework for generative models in offline MBO by finding validation metrics that correlate well with the ground truth oracle
*) apply the framework to diffusion models to find that the agreement metric is most predictive of the test score and that the classifier guidance strength is a most crucial parameter

**Audience:**

No

**Claims And Evidence:**

No

**Requested Changes:**

In my opinion the paper needs to be significantly rewritten so that a broad set of readers can follow and it needs to clearly demonstrate the value of its contributions.

Furthermore, several typos need to be fixed. For example:
-	Before Eq. (2), there are unclosed parentheses in D
-	Inconsistent notation in Eq. (7), the plus should be a comma
-	Paragraph after Eq. (9): there is an incomplete sentence
-	Sec. 2.3, third sentence: which is computed ON the held-out validation set
-	Algorithm 1: “values and stored” -> are stored
-	Sec 4.1: “and no we did not obtain favourable results” – no?

**Strengths And Weaknesses:**

I had a hard time reading the paper and trying to assess the value of its contributions, and I assume I did not properly understand several parts. At first I thought that would be on me and I should dive deeper, but after looking at the code (which the authors claim to be a contribution), I don't have much motivation to do so anymore: the code doesn't come with any (!) instructions or comments how to run it / set it up, and there are also no comments in any of the files explaining what the files are supposed to do.

It feels the authors made similar efforts when trying to make their paper accessible to a broad set of readers (that do not necessarily have prior knowledge about model-based optimization or the Design Bench) and to clearly explain / show the value of the paper's contributions:
*) in the experiments section, why is there no discussion of / comparison to a baseline approach that just evaluates f?
*) Page 5, before Eq. (11): what is y in D_{train}?
*) in the literature on classifier-based guidance, y is a class label, but here it's a score
*) Page 5, Eq. (3): expectation and a minus is missing – cf. Eq. (3) from Ho et al. (2020)
*) In Section 2.3, the framework is designed "to be consistent with the evaluation setup of Design Bench", but the Design Bench is only introduced in Section 3.
*) While the sketches and tables could be helpful, their captions are not self-contained. For example, what does 'min/max?' mean in Table 1 or what is the purpose of the last step in Figure 2.

---

> ### Author Response · Authors · 2023-10-24
> **Reply**
>
> > It feels the authors made similar efforts when trying to make their paper accessible to a broad set of readers (that do not necessarily have prior knowledge about model-based optimization or the Design Bench) and to clearly explain / show the value of the paper's contributions:
>
> We thank you for taking the time to go through the paper, especially considering the difficulties you have mentioned. It is indeed a difficult balance between introducing lots of new concepts which are important (i.e. the way we perform evaluation, statistical language, the use of diffusion models) and also having a page limit which is reasonable and respects the reviewer's time. We really do appreciate the time you've taken to help us write a clearer paper.
>
> We have updated the entire introduction section to make it even clearer what it is we're trying to do and why it is important, but we will also paraphrase it here with some shorter text. Understanding this paper really boils down to thinking about what the difference is between a typical training/evaluation setup in ML versus what it is in MBO:
>
> In a traditional setup training and evaluation is straightforward, because your test set comes from the same distribution as your training set and the test set already has labels from the ground truth. This is true for almost all machine learning pipelines, e.g. evaluating classifiers, generative models, etc. Therefore, whatever metric that is computed on the test also has access to those ground truth labels (e.g. the mean log likelihood over the test set).
>
> In MBO, we want to use our generative model to generate new examples, i.e. sample $x \sim p_{\theta}(x|y)$ for large $y$'s', since we want to sample high reward examples. We can think of this as generating our own 'synthetic' test set. However, we don't know the true labels for those examples and we would need to evaluate the ground truth oracle $f$ for each example in our 'fake test set', but this is expensive since evaluating $f$ means executing a real world process (e.g. synthesising a molecule in a wet lab). The alternative is to approximate $f$ with $\tilde{f}$ and use that to evaluate our synthetic test set, but this is flaky from an evaluation point of view.
>
> This is an inevitable problem in offline MBO, but we ask ourselves whether we can make this process more reliable by carefully choosing the right validation metric, one that is well correlated with the ground truth. Since we assume validation metrics are cheap to compute, they can be evaluated many times over without being uneconomical.
>
> > in the experiments section, why is there no discussion of / comparison to a baseline approach that just evaluates f? *)
>
> The baseline approach is quoted from Design Bench in Table 3, these are the entries in the table starting with "Grad". We renamed these in the new draft to "Grad ascent", "Grad ascent (min)", and "Grad ascent (ensemble)", respectively. The details regarding these baselines can be found in Section 6 of Design Bench (Trabucco et al).
>
> Furthermore, Design Bench's "COMs" model can be thought of as a highly regularised form of $f_{\theta}$, and therefore can be thought of as a strong baseline.
>
> > Page 5, before Eq. (11): what is y in D_{train}?
> > *) in the literature on classifier-based guidance, y is a class label, but here it's a score
>
> We apologise for the confusion, this seems to be side effect of the fact that for DDPM generative models they also use the word 'score' which has a specific meaning, in particular $\nabla_x \log p(x)$, whereas we have been consistently using it refer to what would also be called the 'reward' $y = f(x)$. We noticed that in the paper we referred to it as the 'label' on one occasion, and this may also confuse readers in the sense that they may automatically think that $y$ is a discrete variable rather than a continuous one, like in the case of regression.
>
> We have uploaded a version of the manuscript where $y$ is exclusively referred to as the _reward_. We also removed the text which referred to it as a class label. To be clear, $y = f(x)$ is a continuous variable and is assumed to be positive.
>
> > *) Page 5, Eq. (3): expectation and a minus is missing
>
> Thank you very much for noting this. We have fixed the missing expectation and flipped the quotient in the RHS.
>
> > *) In Section 2.3, the framework is designed "to be consistent with the evaluation setup of Design Bench", but the Design Bench is only introduced in Section 3.
>
> Thank you, we have fixed this and simply said that the scheme we have used is designed to be consistent with Trabucco et al, without making explicit reference to Design Bench.

---

> ### Author Response · Authors · 2023-10-24
> **Reply continued**
>
> > *) While the sketches and tables could be helpful, their captions are not self-contained. For example, what does 'min/max?' mean in Table 1
>
> The min/max in the table was meant to distinguish between validation metrics that should be either maximised or minimised, however we decided to remove this column due to the aforementioned confusion. Since `M_score` (now renamed to `M_reward`, see previous reply) and `M_dc` are metrics that should be maximised, we instead prefix them with a negative sign.
>
> > ... or what is the purpose of the last step in Figure 2.
>
> We have updated the Figure 2 image to make it more clear how it relates to the rest of the text. We have also re-written the caption accordingly.
>
> ---
>
> > the code doesn't come with any (!) instructions or comments how to run it / set it up, and there are also no comments in any of the files explaining what the files are supposed to do.
>
> Hi, we apologise for this inconvenience. The original intention was to have a fully documented and reproducible Github repository timed with acceptance  and official publication of the paper. The code release associated with the submission itself was more intended as a reference material in case reviewers were curious about specific implementation details.
>
> If there are any remaining details in the paper which are unclear, please do not hesitate to let us know. Your feedback has been very helpful so far with helping us smooth out the text.

---

### Author Response · Authors · 2024-05-26
**Updates for camera ready**

Hi Action Editor,

Thank you for your recommendation and suggestions to further improve the the paper. To answer the aforementioned questions or comments:

> Regarding inconclusive results, I read the paper and I can't quite tell from the discussion whether the point of Table 3 is to showcase the superiority of the approach compared to the baseline approaches, or to provide some form of insight into the framework.

Yes, in retrospect a baseline from Design Bench side (e.g. COMs, MINs) would have been ideal to add to the table so that it could be directly compared to the its result on the Design Bench side (which wouldn't have a validation set). On the positive side however, we have shown from figures in the paper that validation metrics are strongly correlated with the ground truth (whether they be -ve or +ve), so we would expect that any baseline from Design Bench to perform better when it's combined with the right validation metric.

> As one reviewer points out, it seems to win in 2 out of 4 cases, although it looks quite competitive in the other 2. How much tuning is required to get to this level of performance?

As far as diffusion model training is concerned, the amount of tuning should be minimal. The most important hyperparameter is actually for inference time which is the classifier-free (or classifier-based) guidance parameter, which controls the diversity / quality trade-off. Note that since we use the DDPM formulation of diffusion (Ho et al, 2020) the sampling algorithm is rather straightforward, i.e. it is just ancestral sampling. But with diffusion models in general there are many sampling algorithms (e.g. see EDM from Karras et al, 2022) and ways to formulate the original training objective which were not explored in this work. This necessitates some future work in exploring the "design space" of generative models but in the context of MBO. We now mention this in the conclusion.

> Will the conclusions hold across generative models and other tasks? That is, the framework is interesting, but it's not clear what the general takeaway of the results are beyond the limited scope of the models/datasets considered. Some discussion on this would be very helpful.

The main takeaway of the paper is in demonstrating that one should be thinking carefully about what needs to be measured during model selection, which in our case is extrapolation. This necessitates having a validation set which is not the same distribution as the training set. However, something needs to be computed over that validation set which leads us to validation metrics and we explore various candidates for it. The takeaway message is not so much a bold claim (i.e. "free lunch" and finding a validation metric to "rule them all", or prematurely extrapolating our findings to the broader literature) but rather bringing attention to an issue which appears to be relatively unexplored. The choice to validate the approach on diffusion models was done because of the recent interest they have attracted in the generative modelling space, but we agree that the nature of the our evaluation framework means that there is room for it to be evaluated against other generative models and datasets to deliver a stronger conclusion.

> The other lingering concern centers around related work, although I do see that you cite and discuss 2/3 of Reviewer qSSg's suggested references. I suggest revisiting their comments and ensuring that you fully discuss related work as per their recommendation.

Thank you, we added a paragraph in Related Work about OptFormers and how it relates to our setting.

> With these in mind, I am still inclined to accept because I believe that the core idea is interesting and that it sufficiently satisfies the requirements for TMLR of supported claims and audience interest.

Thank you again for finding our work interesting and of value to the community.

---

### Decision · Action_Editor_vSMx · 2024-04-04

**Recommendation:** Accept with minor revision

**Comment:**

The primary concerns around the initial version were writing quality, clarity, and organization. This has been largely addressed in the second revision. However, two of the reviewers still lean towards reject, citing inconclusive results and missing related work.

Regarding inconclusive results, I read the paper and I can't quite tell from the discussion whether the point of Table 3 is to showcase the superiority of the approach compared to the baseline approaches, or to provide some form of insight into the framework. As one reviewer points out, it seems to win in 2 out of 4 cases, although it looks quite competitive in the other 2. How much tuning is required to get to this level of performance? Will the conclusions hold across generative models and other tasks? That is, the framework is interesting, but it's not clear what the general takeaway of the results are beyond the limited scope of the models/datasets considered. Some discussion on this would be very helpful.

The other lingering concern centers around related work, although I do see that you cite and discuss 2/3 of Reviewer qSSg's suggested references. I suggest revisiting their comments and ensuring that you fully discuss related work as per their recommendation.

With these in mind, I am still inclined to accept because I believe that the core idea is interesting and that it sufficiently satisfies the requirements for TMLR of supported claims and audience interest.

**Audience:**

Yes, I believe that there will be individuals in the MBO community who will find this interesting.

**Claims And Evidence:**

The paper doesn't make any bold claims about results, rather it proposes a new framework for thinking about offline MBO where the true objective is prohibitively expensive and the best we can do is to measure progress on an extrapolative validation set. The paper takes a generative view of the problem and it provides a suite of experiments exploring different validation metrics using diffusion (with classifier-based and classifier-free guidance) models as the surrogate.

The paper could have gone further and investigated a different family of generative models, or a different domain of tasks outside of Design Bench. In its current form, it is unclear if the lessons learned on this set of validation metrics will necessarily hold true in other domains. However, for its given scope it is comprehensive and I therefore lean on the positive side for this.